# Transport of FNPP1-derived radiocaesium from subtropical mode water in the western North Pacific Ocean to the Sea of Japan

Yayoi Inomata[1], Michio Aoyama[2], Yasunori Hamajima[1], Masatoshi Yamada[3]

[1]Institute of Nature and Environmental Technology, Kanazawa University, Kanazawa, 920-1156, Japan
[2]Institute of Environmental Radioactivity, Fukushima University, Fukushima, 960-1296, Japan
[3]Institute of Radiation Emergency Medicine, Hirosaki University, Hirosaki, 036-8564, Japan

*Correspondence to*: Yayoi Inomata (yinomata@se.kanazawa-u.ac.jp)

**Abstract.**

The $^{137}$Cs activity concentration in the Sea of Japan (SOJ) has been increasing since 2012/2013 and reached a maximum in 2015/2016 of approximately 3.4 Bq m$^{-3}$, while the pre-Fukushima accident $^{137}$Cs activity concentration was approximately 1.5 Bq m$^{-3}$. It was also observed that the $^{134}$Cs/$^{137}$Cs activity ratios ranged from 0.36 to 0.51 in 2016. Taking into account that the radioactive decay and ocean mixing, these $^{134}$Cs/$^{137}$Cs activity ratios are evidence that the increase in the $^{137}$Cs activity concentration was derived from the Fukushima accident. In the North Pacific South of Japan (NPSJ), the higher $^{137}$Cs activity was observed in the 300 m depths, which corresponds to the subtropical mode water (STMW) based on the potential density, in 2012/2013. In the East China Sea (ECS), the clear increase in the $^{137}$Cs activity concentration started at 140 meters depth, density=25.2 kgm$^{-3}$, in April 2013, and propagated to the surface layers at approximately 0-50 meters depth, and showing a maximum in 2015 and decreasing in the following years. In the ECS, the maximum Fukushima-radiocaesium activity concentration in surface water was observed in 2014/2015, while it in the SOJ was observed in 2015/2016. Therefore, the propagation of Fukushima-derived radiocaesium in the surface seawater was approximately one year from ECS into the SOJ. These temporal changes in the $^{137}$Cs activity concentration and $^{134}$Cs/$^{137}$Cs activity ratio indicated that a part of the $^{137}$Cs and $^{134}$Cs derived from the Fukushima accident (FNPP1-$^{137}$Cs and FNPP1-$^{134}$Cs) was transported to the ECS and then to the SOJ from the STMW in the NPSJ within several years. The integrated amount of FNPP1-$^{137}$Cs that entered the SOJ until 2016 was estimated to be 0.21±0.01 PBq, which corresponds to 5.0 % of the estimated total amount of FNPP1-$^{137}$Cs in the STMW. The integrated amount of FNPP1-$^{137}$Cs that returned to the North Pacific Ocean through the Tsugaru Strait was estimated to be 0.09±0.01 Bq, which corresponds to 43 % of the total amount of FNPP1-$^{137}$Cs transported to the SOJ and 2.1 % of the estimated total amount of FNPP1-$^{137}$Cs in the STMW.

## 1 Introduction

The Fukushima Daiichi Nuclear Power Plant (FNPP1) accident in March 2011 released radiocaesium ($^{137}$Cs (T$_{1/2}$ of 30.07 yr) and $^{134}$Cs (T$_{1/2}$ of 2.06 yr)) by atmospheric release to the air and the direct discharge of contaminated water to the ocean, mostly in March and April 2011, and 80 % of atmospheric released radiocaesium deposited ocean surface (Aoyama et al., 2016a; Buesseler et al., 2016; Hirose, 2016; Tsumune et al., 2013). The $^{137}$Cs activity concentration in the surface seawater of the North Pacific Ocean after the FNPP1 accident ranged from a few Bq m$^{-3}$ to ca. 1 kBq m$^{-3}$ (e.g., Aoyama et al., 2012, 2013; Honda et al., 2012; Kaeriyama et al., 2013, 2014). A basin-scale measurement indicated that the FNPP1-derived radiocaesium was observed in a region from 25°N to 50°N and from 135°E to 135°W in April and May of 2011 (e.g., Aoyama et al., 2013; 2016a; Inomata et al., 2016; Tsubono et al., 2016). The radiocaesium moved eastward to approximately 40°N with the North Pacific current systems, such as the Kuroshio and Kuroshio Extension, at a rate of approximately 8 cm sec$^{-1}$ and reached 165°E in July-September 2011 and the 180$^{th}$ meridian in January-March 2012 (Aoyama et al., 2012, 2013). FNPP1-derived radiocaesium was also found further south in the North Pacific Ocean, with a subsurface maximum at a depth of approximately 300 m (Kaeriyama et al., 2013, 2014, 2016; Kumamoto et al., 2013, 2017). These subsurface maxima of $^{137}$Cs were found in the subtropical mode water (STMW; potential density ($\sigma_\theta$) =25.0-25.6 kgm$^{-3}$; Masuzawa,

1969) and central mode water (CMW; $\sigma_\theta$=26.0-26.5 kgm$^{-3}$) (Aoyama et al., 2016a; Kumamoto et al., 2014, 2015). It was deduced that atmospherically-deposited $^{137}$Cs south of the Kuroshio Extension subducted into the STMW and transported southward within 10 months after the FNPP1 accident.

Ocean circulated models captured that the FNPP1-derived radiocaesium were transported in the North Pacific Ocean with advecting and diluting in the surface seawater (Tsubono et al., 2016; Rossi et al., 2014). It was also revealed that dilution due to horizontal and vertical dispersion in the vicinity of Kuroshio leaded to a rapid decrease of the radiocaesium activity concentrations (Behrens et al., 2012). Rossi et al. (2013) reported that simulated FNPP1-derived $^{137}$Cs were penetrated into the ocean interior by intense subduction and vertical mixing on winter season around the formation regions of mode waters as described by Oka and Qiu (2012). In their results, the higher $^{137}$Cs activity concentrations associated by the subduction was also reproduced in the STMW, Denser CMW, and Lighter CMW. About 43% of FNPP1-$^{137}$Cs was transported downward below the mixed layer by eddy processes (Kamidaira et al., 2018).

$^{137}$Cs was injected into the environment by the large-scale atmospheric nuclear weapons testing that occurred in the late 1950s and early 1960s and the Chernobyl accident in 1986, therefore before the FNPP1 accident, $^{137}$Cs existed in the North Pacific Ocean and its marginal seas (Aoyama et al., 2006; Inomata et al., 2009). Furthermore, the dumping of radioactive wastes by Russia and the former USSR occurred in the northern part of the Sea of Japan (SOJ), although this caused no significant increase in the activity concentrations (Miyao et al., 1998). In the 2000s, the $^{137}$Cs activity concentrations had an almost homogenous distribution in the Pacific Ocean, although relatively high activity was observed in the western part of the subtropical gyre in the North Pacific Ocean (exceeding 2 Bq m$^{-3}$) and the South Pacific Ocean (exceeding 1.5 Bq m$^{-3}$) (Aoyama et al., 2012). In March-April 2011, FNPP1-derived $^{137}$Cs and $^{134}$Cs (FNPP1-$^{137}$Cs, FNPP1-$^{134}$Cs) were released into the North Pacific Ocean. There have been two sources of $^{137}$Cs in the North Pacific Ocean and its marginal seas, the $^{137}$Cs released by global fallout (global fallout-$^{137}$Cs) and FNPP1-$^{137}$Cs. In contrast, the $^{134}$Cs derived from global fallout and the Chernobyl accident decayed out at the measurement in 1993 due to its shorter lifetime (Miyao et al., 1998). Therefore, $^{134}$Cs is regarded as an adequate chemical tracer of the radiocaesium derived from the FNPP1 accident.

In the northeastern part of the SOJ, atmospherically deposited radiocaesium were caused to approximately 1-2 times higher activity concentrations of radiocaesium in May 2011 (1.5-2.8 Bqm$^{-3}$) in northeastern SOJ than those of before the FNPP1 accident (~1.5 Bqm$^{-3}$) (Fig. 1). By the end of 2011, the $^{137}$Cs activity concentrations in the northeastern part of the SOJ had rapidly decreased to almost the same levels as those before the FNPP1 accident (Inoue et al., 2012). However, several sets of seawater monitoring data have indicated an increase in the $^{137}$Cs activity concentration after 2012.

The SOJ is located between the Eurasian continent and the Japanese archipelago. The area is 1008000 km$^2$, and the mean depth is 1667 m (Menard and Smith, 1966). The SOJ is connected to the East China Sea at its southwest through the Tsushima Straits and connected to the North Pacific Ocean at its northeast through the Tsugaru Straits. Warm and saline seawater passes through the Tsushima Strait as the Tsushima Warm Current (TWC), and this current splits into mainly two paths: One is the nearshore current along the west coast of Honshu Island, Japan, and the seawater passes through the Tsugaru Straits and enters to the Pacific Ocean again. The seawater transported to the northward passes thought the Soya Strait and connected to the Okhotsk Sea. Another current flows north of the Korean Peninsula. This current meets the North Korean Cold Current, which is the prolongation of the Liman Cold Current. Therefore, this northward warm subtropical waters and southward cold subarctic waters form the Polar Front meet at approximately 40°N, and the SOJ is largely divided into two regions (Prants et al., 2015). The increased FNPP1-$^{137}$Cs radioactivity concentrations were observed in the west coast of Honshu island (Aoyama et al., 2017).

Aoyama et al. (2017) reported the increased of $^{137}$Cs activity concentrations in the surface seawater around the Japanese islands and ECS in winter 2015/2016 as a preliminary result. This research was investigated the temporal variation of $^{137}$Cs activities after the 1960's in the SOJ, and estimated the FNPP1-derived $^{137}$Cs activity concentrations after the year of 2011. We also investigated the vertical distributions of the $^{137}$Cs activity concentrations in the NPSJ, ECS, and SOJ. We also

discussed about FNPP1-[137]Cs transport route from the North Pacific South of Japan (NPSJ) to the SOJ, estimation of transport amount with uncertainty of FNPP1-[137]Cs into the SOJ as well as return amount of FNPP1-[137]Cs into the North Pacific Ocean during the period from 2012 to 2016.

## 2 Sampling, measurements and methods

### 2.1. Data sources

After the FNPP1 accident, many radiocaesium measurements were taken in the SOJ and western North Pacific Ocean (Fig. 1). To elucidate the temporal and spatial distributions of the radiocaesium activity concentrations, we use as many data points as possible. We, therefore, compiled all available data from the literature and reported studies. Most of the data before the FNPP1 accident was included in the database, "Historical Artificial Radionuclides in the Pacific Ocean and its Marginal Seas (HAM database)" (Aoyama and Hirose, 2003 and their updated version). The data observed after the FNPP1 accident were shown in Aoyama et al. (2016a). The term "surface seawater" used in this study defines a sample collected shallower than 10 m depth.

We also focused on the Japanese government's monitoring data at Tomari (42.98-43.17°N, 140.21-140.30°E), Aomori (41.13-41.22°N, 141.50-141.67°E), Niigata (37.62-38.10°N, 138.38-138.84°E), Ishikawa (36.87-37.29°N, 136.43-136.47°E), Fukui (35.75-36.09°N, 135.50-135.83°E), Shimane (35.67-35.80°N, 132.87-133.2°E), Saga (33.57-33.62°N, 129.73-129.98°E), and Kagoshima (31.58-31.93°N, 130.02-130.15°E) (Marine Ecology Research Institute, 2011, 2012, 2013, 2014, 2015, 2016) (Fig. 1). These measurements were taken once a year (from the middle of May to early June). Near the Aomori sites, offshore monitoring was also conducted twice a year (May and October) at the Rokkasho Reprocessing Plant (39.5-41.4°N, 141.5-142.3°E). Seawater was sampled from 0-664 m with different depths at each monitoring site. Monitoring data (304-01(33.0°N, 127.7°E), 105-11(37.3°N, 131.3°E)) from the Korean government was also used in this analysis (Korea Institute of Nuclear Safety, 2011, 2012, 2013, 2014, 2015, 2016) (Fig. 1). At these monitoring sites, the surface seawater (0 m) measurements were taken four times (February, April, August, October) a year.

### 2.2. Measurement of the [134]Cs activity concentration

Because the [134]Cs activity concentrations in most of the monitoring datasets were less than the detection limit due to low measurement capabilities, we also collected seawater samples to investigate the [134]Cs activity concentrations. The sample volume ranged from a few litres to ten litres. The radiocaesium in the sampled seawater was extracted by the ammonium phosphomolybdate (AMP)/Cs compound method improved by Aoyama and Hirose (2008b). The AMP/Cs compound was measured by using ultra-low-background gamma-ray detectors in the Low Level Radioactivity Concentration Laboratory, Kanazawa University (Hamajima and Komura, 2004) to measure [134]Cs and [137]Cs activity concentrations. The measurement details are described in Aoyama and Hirose (2008b) and Aoyama et al. (2016a). All radioactivity concentrations shown in this research were decay corrected at the time of sample collection. In addition, we also determined the [134]Cs/[137]Cs activity ratio decay corrected to 11 March 2011 at the time of the FNPP1 accident.

### 2.3. Estimation of the FNPP1-[137]Cs activity concentration

In order to investigate the contribution of FNPP1-[137]Cs, increased [137]Cs activity concentrations derived from the FNPP1 accident were estimated in this study. As shown in Fig. 2., the [137]Cs activity concentrations in the SOJ exponentially decreased, and the decreasing rate from 2000 to 2010 become to slow. By assuming that the apparent half-residence time of the global fallout-[137]Cs was approximately the same before and after the FNPP1, the global fallout-[137]Cs activity concentrations were estimated as shown in Fig. SI1. Then, the FNPP1-[137]Cs was estimated from the difference between the measured and the extrapolated value of the half-year average in the SOJ during the period from 2000 to 2010 (Fig. SI1).

**2.4. Estimation of the [137]Cs transport amounts during the period from 2012 to 2016**

We estimated the transported amounts of FNPP1-[137]Cs in each region during the period from 2012 to 2016 by using the following equation (1):

$$\text{Transport amount} = \sum_{i=2012}^{n} \left[\text{FNPP1-}^{137}\text{Cs activity concentration in year i}\right] \times$$

$$\left[\text{annual average seawater transport volume in year i}\right] \quad (1)$$

where n = 2016, the annual average FNPP1-[137]Cs activity concentrations at the station Saga and the station 304-11 were used for the ECS, the FNPP1-[137]Cs activity concentrations at the station Shimane were used for the eastern TWC, the FNPP1-[137]Cs activity concentrations at the station 105-01 were used for the western TWC, and the FNPP1-[137]Cs activity concentrations at the station Aomori were used for the transported amount to the North Pacific Ocean via the Tsugaru Strait. The amount transported FNPP1-[137]Cs northward along the western coast of Hokkaido was estimated from the difference between the total inflow of FNPP1-[137]Cs at the entrance and the outflow that passed through the Tsugaru Strait. The annual average volume of seawater transported through the TWC was used the data obtained from Fukudome et al. (2010) and Han et al. (2016). The annual average seawater transport volumes of the TWC into the SOJ ranged from $2.59\pm0.50 \times 10^6$ m$^3$ s$^{-1}$ to $2.82\pm0.58 \times 10^6$ m$^3$ s$^{-1}$, of which the annual average seawater transport volume through the east and west channels of the Tsushima Strait were $1.08\pm0.26$-$1.26\pm0.27 \times 10^6$ m$^3$ s$^{-1}$ and $1.43\pm0.38$-$1.56\pm0.47 \times 10^6$ m$^3$ s$^{-1}$ during the period from 2012 to 2016, respectively. The volume transported through the Tsugaru Strait was estimated at $1.0$–$1.3 \times 10^6$ m$^3$ s$^{-1}$ by Nishida et al. (2003), and this was used because we did not have *in situ* monitoring data for the seawater transport volume. The evaluation of seasonal variations was difficult in this study because of the limited monitoring data, which were measured once a year at the Japanese monitoring site and four times a year at the Korean monitoring sites.

**3 Results**

**3.1 Increased [137]Cs activity concentrations in the ECS and the SOJ**

Fig. 2 shows the temporal variation in the [137]Cs activity concentrations in the surface seawater of the SOJ after the 1960s. The half-year average values of the [137]Cs activity concentrations were also plotted. The [137]Cs activity concentrations exponentially decreased from 1960-1970; these concentrations originated from atmospheric nuclear weapons testing (Aoyama, 2009). The increase in the [137]Cs activity concentrations in 1986 was derived from the Chernobyl accident (Miyao et al., 1998). The apparent half-residence time during the period from 1970 to 1990 was estimated to be $16.3\pm0.5$ years (Inomata et al., 2009). In the 2000s, the decreasing rate of [137]Cs activity concentrations were small and the [137]Cs activity concentrations in the surface seawater reached low, 1.5-2.4 Bq m$^{-3}$ as for half-year average value. Just several months after the FNPP1 accident, high activity concentrations of [137]Cs, up to 3.3 Bq m$^{-3}$, were observed. As shown in Fig. 1, higher [137]Cs activity concentrations were observed in the northeastern part of the SOJ than those in the southwestern part of the SOJ. These high [137]Cs activity concentrations were associated with the wet deposition of [137]Cs released into the atmosphere during the FNPP1 accident in March-April 2011. Other than the increase in the [137]Cs activity concentrations just after the FNPP1 in 2011, the [137]Cs activity concentrations gradually increased in the SOJ, and the [137]Cs activity concentrations in 2016 reached 2.6 Bq m$^{-3}$, which is the same level as those in 1998.

Fig. 3 shows the FNPP1-[137]Cs activity concentrations measured in the ECS and SOJ by the Japanese and Korean governments. These monitoring stations are located along a branch of the Kuroshio in the ECS and along the western and eastern TWC in the SOJ. The long-term variations in the [137]Cs activity concentrations and FNPP1-[137]Cs at each monitoring station are also shown in Fig. SI2-9. Excluding the high activity concentrations at Aomori in 2011 caused by the wet deposition of FNPP1-[137]Cs released to the atmosphere, an increase in the FNPP1-[137]Cs in the SOJ started in 2012/2013, and the FNPP1-[137]Cs gradually increased up to 1.3 Bq m$^{-3}$ until 2016 (Fig. 3a, b). The lower activity concentrations at Niigata (Fig. 3b) resulted from the measurements at another station near the Sado Island, off Niigata. At these monitoring stations, there was no significant decreasing trend of the FNPP1-[137]Cs activity concentrations in 2015 and 2016. On the other hand,

the FNPP1-[137]Cs activity concentrations at stations located upstream from the SOJ, in the ECS (Fig. 3c; Kagoshima, Saga, and station 314-01), also increased from 2012/2013. Particularly, the increase was clearly observed after 2014 in the ECS. We should note that the FNPP1-[137]Cs activity concentrations in 2016 tended to slightly decrease compared to those in 2015. At the station 105-11, which is located off South Korea (Fig. 1), marked increases in the FNPP1-[137]Cs activity concentrations were observed in 2014, and the FNPP1-[137]Cs activity concentrations reached approximately 1.4 Bq m$^{-3}$ in 2015 (Fig. 3d). In 2016, the FNPP1-[137]Cs concentrations were decreased relative to those in the preceding years.

The [134]Cs/[137]Cs activity ratios in 2016 ranged from 0.27 to 0.51 (Fig. 4). There was no significant difference in the [134]Cs/[137]Cs activity ratio among the stations in the ECS and the SOJ.

Fig. 5 show the temporal variation in the [137]Cs activity concentrations at different depths at station 314-01 in the ECS (Fig. 5a; 0-140 m) and station 105-11 in the SOJ (Fig. 5b; 0-2000 m). The increase in the [137]Cs activity concentrations at the subsurface layer (140 m at the station 314-01, 200 m at the station 105-11) occurred in 2013 and this was approximately 1 year earlier than that in the shallower layers. At station 314-01, the [137]Cs activity concentrations at 140 m depth were higher than those at 0 m depth beginning in 2013, and the [137]Cs activity concentrations increased up to 3.2±0.28 Bq m$^{-3}$ in 2014 and then tended to decrease after 2015. At shallower depths (0-50 m), increased [137]Cs activity concentrations were found in 2014, and there was no significant decrease in the [137]Cs activity concentrations in 2015 and 2016. At the station 105-11, the increase in the [137]Cs activity concentrations started earlier in the subsurface seawater (200 m depth) in 2013. In 2014, the [137]Cs activity concentrations in surface seawater also increased and reached 2.8±0.2 Bqm$^{-3}$ in 2015. Decreases in the [137]Cs activity concentrations at 200 m and 0 m depth were observed after 2015, and a subsurface peak in the [137]Cs activity concentration at 200 m depth was not observed after 2015. Similar variations, in which the increase in the [137]Cs activity concentration started from the ocean interior, were observed at the Kagoshima and Saga monitoring stations (Fig. SI10).

**3.2 Propagation of radiocaesium from the upstream region (NPSJ and ECS) to the downstream region (the SOJ)**

The vertical distributions of the [137]Cs activity concentration over the depth and potential density ($\sigma_\theta$, kg m$^{-3}$) profiles in the North Pacific South of Japan (NPSJ), in the ECS, and at the station 105-11 are shown in Fig. 6. In the NPSJ, the [137]Cs activity concentrations showed subsurface maxima at approximately 300 m depth in 2012-2013, with activities of 8.2-12.3 Bqm$^{-3}$ as shown in Fig. 6a. These high [137]Cs activity concentrations were measured in the region between 136-138°E and 26-30°N. After 2014, a subsurface peak in the [137]Cs activity concentration was not observed. These subsurface peaks in the [137]Cs activity concentrations were found in the layer corresponding to a potential density of 25.2 kg m$^{-3}$ (Fig. 6b). This is consistent with previous findings of an [137]Cs activity maximum in STMW with a potential density of 25.0-25.6 kg m$^{-3}$ (Kaeriyama et al., 2014; Kumamoto et al., 2014). In the ECS, the [137]Cs activity concentrations gradually increased beginning in 2012 and attained the maximum activity concentrations (2.9±0.24 Bqm$^{-3}$) in 2015, following by a decreasing trend in the [137]Cs activity concentrations in 2016, as shown in Fig. 6c. The higher [137]Cs activity concentrations above 2 Bq m$^{-3}$ in the ECS were found in the layer corresponding to a potential density of 23.6-25.2 kg m$^{-3}$, as shown in Fig. 6d. On the other hand, the [137]Cs activity concentrations at the station 105-11, located in the western SOJ, decreased with increasing depth until 500 m (Fig. 6e). The higher [137]Cs activity concentrations at the station 105-11 were measured in 2014 and 2015, and the [137]Cs activity concentrations decreased in 2016. The [137]Cs activity concentrations above 2 Bq m$^{-3}$ at the station 105-11, except one sample measured at 2015, were located in the layer with a potential density of 25.8-27.1 kg m$^{-3}$ (Fig. 6f).

Fig. 7 displays the Hovmoller diagrams of the [137]Cs activity concentrations at potential temperatures of 25.2±0.5 kgm$^{-3}$ along the TWC in the ECS and coastal site of eastern SOJ. These potential density surfaces were selected to show the maximum [137]Cs activity concentrations observed in the ECS. The vertical distributions of the [137]Cs activity concentrations with depth and potential density at each monitoring station are displayed in Fig. SI11. Note that in the SOJ, the vertical distributions of the [137]Cs activity concentrations below 250 m were almost constant, and the subsurface peak of [137]Cs was not found at the monitoring stations along the eastern TWC (Fig. SI11). It is noted that the [137]Cs activity concentrations before

the FNPP1 accident were approximately 1.5 Bq m$^{-3}$. As shown in Fig. 7, in the ECS, the $^{137}$Cs activity concentrations gradually increased and attained the maximum in 2014/2015. The $^{137}$Cs activity concentrations in the ECS tended to decrease in 2016 in a layer with a density of 25.2±0.5 kgm$^{-3}$. In the southwestern part of the SOJ (Shimane, Fukui, Ishikawa, Niigata), the $^{137}$Cs activity concentrations also gradually increased beginning in 2012, and the activity concentrations attained a maximum of 2.5 Bq m$^{-3}$ in 2015/2016; these trends were almost the same as those at the monitoring stations in the ECS. In the northwestern SOJ (Aomori and Tomari), the $^{137}$Cs activity concentrations were slightly increased and higher activity concentrations above 2 Bq m$^{-3}$ were observed in 2016. These results revealed that the propagation of FNPP1-$^{137}$Cs occurred within one year from the ECS (32-34°N) to the SOJ (35-38°N) along the TWC as shown in Fig. 7.

**3.3. Transport process and total amount of FNPP1- $^{137}$Cs transported into the SOJ during the period from 2012 to 2016**

We describe the possible pathway of FNPP1-$^{137}$Cs from the Pacific Ocean to the SOJ by ECS. Taking into account the physical ocean circulation, FNPP1-$^{137}$Cs in the STMW would be transported to southwestward by the westward undercurrent as reported by Oka (2009). The STMW containing $^{137}$Cs would reach the western boundary at lower latitudes, is entrained into the Kuroshio (Oka and Qiu, 2012), and is northward transported to the west of Kyushu by the TWC bifurcated from the Kuroshio. Then, FNPP1-$^{137}$Cs was entered the SOJ through the Tsushima Strait by the TWC. FNPP1-$^{137}$Cs was divided into western (coastal station in Korea) and eastern (eastern coast of the Japanese Islands) portions along with the TWC. In addition, the remaining FNPP1-$^{137}$Cs in the eastern TWC was transported northward to the western coast of Hokkaido (Fig. 8). Table 1 shows the total amount of FNPP1-$^{137}$Cs during the period from 2012 to 2016. The integrated FNPP1-$^{137}$Cs in the ECS during the period from 2012 to 2016 was estimated to be 0.21±0.01 PBq, which corresponds to 5.0 % of the FNPP1-$^{137}$Cs in the STMW (4.2±1.1 PBq) estimated by Kaeriyama et al. (2016). Connected with the separation of the TWC, the transported amount of FNPP1-$^{137}$Cs in the eastern channel was estimated to be 0.11±0.01 PBq, which corresponds to 2.6 % of the FNPP1-$^{137}$Cs into the STMW. On the other hand, the transported amount in the western channel was estimated to be 0.09±0.01 PBq, which corresponds to 2.1 % of the FNPP1-$^{137}$Cs injected into the STMW. The eastern and western outflow amounts of FNPP1-$^{137}$Cs were consistent with the range of uncertainty for the inflow amounts, 0.21±0.01 PBq. A small part of the transported FNPP1-$^{137}$Cs (0.09±0.01 PBq; 2.1 % of FNPP1-$^{137}$Cs in STMW) passed through the Tsugaru Strait and has already returned to the North Pacific Ocean. In other words, the amount of FNPP1-$^{137}$Cs that has returned to the North Pacific Ocean corresponds to 43 % of the total amount of FNPP1-$^{137}$Cs transported to the SOJ and 2.1 % of the estimated total amount of FNPP1-$^{137}$Cs in the STMW. The remaining 0.03±0.002 PBq (14% of the transported FNPP1-$^{137}$Cs in SOJ; 0.7 % of the transported FNPP1-$^{137}$Cs in STMW) should be transported to the northern part of the SOJ (west of Hokkaido) or be partially transported to the deep region associated with deep convection and surface mixing.

**4    Discussion**

**4.1. Signature of FNPP1-$^{137}$Cs inflow in the SOJ by using $^{134}$Cs/$^{137}$Cs activity ratio**

According to the atmospheric model simulations, atmospheric deposition of FNPP1-$^{137}$Cs occurred over a wide region in the western North Pacific Ocean (30-55°N, 140°E-180°) (e.g., Aoyama et al., 2016). One of the region in which $^{137}$Cs was deposited in south of Kuroshio and Kuroshio Extension regions corresponded to the STMW formation region (Aoyama et al., 2016; Oka et al., 2012, 2013). Because the release of FNPP1-radiocaesium in the atmosphere occurred at the end of March and early April, the deposition of FNPP1-$^{137}$Cs into the North Pacific Ocean was regarded as a point source associated with time. This means that FNPP1 radiocaesium is a very useful tracer to investigate the transport in the North Pacific Ocean. As shown in Fig. 6, the vertical distributions of the $^{137}$Cs activity concentrations in the NPSJ indicate that FNPP1-$^{137}$Cs included in the STMW.

There were found several signatures that (i) The maximum $^{137}$Cs activity concentrations were observed in the subsurface layer in comparison with those in the surface seawater, (ii) This higher $^{137}$Cs activity concentrations were observed in the STMW based on the potential density data, (iii) In the ECS, the potential density in $^{137}$Cs existed seawater was almost same value those in the STMW, suggesting the signature of FNPP1-$^{137}$Cs transport into the ECS from the STMW in the NPSJ, (iv) The $^{137}$Cs activity concentrations in the northern ECS (>30°N) were higher than those in the southern ECS (<30°N) (Fig. SI12). In this study, however, the data was not enough to clarify the transport route in more detail.

Fig. 9 shows the latitudinal distribution of the $^{134}$Cs/$^{137}$Cs activity ratio, which was decay-corrected to 11 March, 2011. The $^{134}$Cs/$^{137}$Cs activity ratio ranged from 0.1 to 0.72, and the ratios at Ogasawara region and in the ECS were almost the same as those in the SOJ (Fig. 9a). Considering that the activity ratio of $^{134}$Cs/$^{137}$Cs that originated from the FNPP1 accident was almost 1 (Busseler et al., 2011), variations in the ratio indicate that FNPP1-labelled seawater mixed with global fallout-labelled seawater (no contamination by FNPP1-$^{137}$Cs) during transport. The highest activity concentration ratio (0.72), which was found near Kagoshima, was a signal of the larger contributions of the FNPP1-radiocaesium. In contrast, a relatively low $^{134}$Cs/$^{137}$Cs activity ratio in the region between 30 and 32°N suggests the small contribution of FNPP1-radiocaesium at the Pacific Ocean site. In the horizontal distributions in the ECS, the $^{137}$Cs activity concentrations tended to increase northward (Fig. SI12). The $^{137}$Cs activity concentrations measured at Okinawa located in the southern ECS (http://search.kankyo-hoshano.go.jp/servlet/search.top?pageSID=113836570), were approximately 1.7±0.47 (~2.8) Bq m$^{-3}$, which are slightly higher rather than those measured before the FNPP1 accident (Fig. SI13). This proves that less FNPP1 radiocaesium was transported in the southern ECS than in the northern ECS as discussed above.

## 4.2. Advection and vertical mixing of FNPP1-$^{137}$Cs in the SOJ

In this study, we revealed that FNPP1-$^{137}$Cs entered the SOJ via the ECS; then, FNPP1-$^{137}$Cs was transported northward with the TWC. In the SOJ, a time lag of the propagation of FNPP1-radiocaesium of approximately one year was observed (Fig. 7). Based on measurements of phosphate, one of the dominant seawater nutrients, Kodama et al. (2016) revealed that the phosphate concentrations in surface seawater during winter were significantly positively correlated with the concentrations in the saline ECS seawater in the preceding summer, and the surface water of the southern SOJ was almost entirely replaced by the ECS seawater during May–October. They also suggested that the transport of water-soluble constituents from the ECS to the SOJ takes at least approximately 0.5 years. The propagation of FNPP1-radiocaesium in the SOJ was consistent with the propagation time scale of nutrients concentration change from the ECS to the SOJ (Kodama et al., 2016).

As shown in Fig. 6e, the $^{137}$Cs activity concentrations at station 105-11, located along the western TWC in the SOJ, were maximum at the surface and gradually decreased with increasing depth. This vertical distribution is different from those in the NPSJ (Fig. 6a) and ECS (Fig. 6c). Particularly, the subsurface peak observed in the NPSJ and ECS did not appear at station 105-11. At station 105-11, most of the $^{137}$Cs existed in seawater with a potential density of 25.7-27.3 kg m$^{-3}$ (Fig. 6f), which was located in a higher potential density layer than that in the NPSJ and ECS. A similar vertical distribution was also observed at the western coast of the Japanese Islands along the eastern TWC (Fig. SI11). These distributions were due to cooling in the surface layer after water was transported from the Tsushima Strait. Physical processes such as the convergence and subduction of surface water inside the eddies are important mechanisms of downward transport of radiocaesium (Miyao et al., 1998; Budyansky et al., 2015). There is a possibility of the seasonality of downward transport of radiocaesium. As expected, the past global fallout-$^{137}$Cs would be already penetrated and accumulated in the deeper layers of the SOJ.

## 4.4. Mass balance of FNPP1-$^{137}$Cs in the SOJ

As for the most reliable estimate, the amount of FNPP1-$^{137}$Cs deposited in the North Pacific Ocean via atmospheric release was 11.7-14.8 PBq (Aoyama et al., 2016b; Tsubono et al., 2016) and the amount of FNPP1-$^{137}$Cs directly released to

the Ocean was 3.5±0.7 PBq (Tsumune et al., 2012, 2013). The FNPP1-$^{137}$Cs inventory in the North Pacific Ocean was estimated to be 15.2-18.3 PBq (Aoyama et al., 2016b), 15.3±2.6 PBq by optimum interpolation analysis (Inomata et al., 2016), and 16.1±1.4 PBq by model simulation (Tsubono et al., 2016). According to the estimation by Kaeriyama et al. (2014), the $^{134}$Cs amount was approximately 4.2±1.1 PBq in the STMW in 2012. Taking into account this estimation by Kaeriyama et al. (2014), the FNPP1-$^{137}$Cs transported into the SOJ from 2012 to 2016 accounted for 5.0 % of the FNPP1-$^{137}$Cs inventory in the STMW. Of this, approximately 60-65 % of the total inflow of FNPP1-$^{137}$Cs in the SOJ was transported in 2015 and 2016. After entering the SOJ, the FNPP1-$^{137}$Cs followed a divergent path. The amounts of FNPP1-$^{137}$Cs transported eastward (2.6 % of the FNPP1-$^{137}$Cs inventory in the STMW) and westward (2.1 % of the FNPP1-$^{137}$Cs inventory in the STMW) along the TWC were similar or slightly larger along the eastern TWC. Of the FNPP1-$^{137}$Cs transported in the SOJ, approximately 43 % returned to the North Pacific Ocean through the Tsugaru Straight during 2012 to 2016.

Since 5 % of the FNPP1-$^{137}$Cs in the STMW was transported into the SOJ, 95 % of that amount might remain in the STMW. Based on long-term measurements of the $^{137}$Cs activity concentration adjacent to the FNPP1, Tsumune et al. (2017) estimated that the direct release rate of FNPP1-$^{137}$Cs was $2.2\times10^{14}$ Bq day$^{-1}$ until November 2011, after which it decreased exponentially with time to $3.9\times10^{9}$ Bq day$^{-1}$ on 26 October 2015. Assuming that the rate of decrease in the $^{137}$Cs activity concentrations remained the same from 26 October 2015 to 31 December 2016, the total amount of $^{137}$Cs directly released from the FNPP1 site during the period from 2012 to 2016 was estimated to be 0.03 PBq (Tsumune et al., 2016). However, not all of the directly released $^{137}$Cs subducted into the STNW, 0.03 PBq corresponds to 0.7 % of $^{137}$Cs inventory in the STMW. We can conclude that the FNPP1-$^{137}$Cs in the STMW was decreasing during the study period.

Based on an analysis of historical data since 1950, the $^{137}$Cs activity concentrations in the western North Pacific Ocean were exponentially decreased. However, the $^{137}$Cs activity concentrations after 2000 have remained almost constant at 1.5-2 Bq m$^{-3}$ (Inomata et al., 2009), which indicates a source of $^{137}$Cs. Considering that the contribution of global fallout-$^{137}$Cs deposition amount was scarcely any after the 1970s, the $^{137}$Cs activity concentrations in the seawater of the North Pacific Ocean were controlled by physical oceanographic processes such as advection and diffusion until before the FNPP1 accident (Inomata et al., 2009). Several studies reported that the subsurface maxima of $^{137}$Cs derived from global fallout existed in layers corresponding to the potential densities of the STMW and CMW (e.g., Aoyama et al., 2008a). A possible explanation for the constant $^{137}$Cs after 2000 is that the $^{137}$Cs subducted in the STMW and CMW in the 1960s might have returned to the surface after a few decades. However, the increase in the FNPP1-$^{137}$Cs activity concentrations in the SOJ derived from the FNPP1 accident began in 2012/2013. This would be existence of the faster transport route from the North Pacific Ocean to the SOJ along with the subtropical gyre.

## 5. Conclusion

The $^{137}$Cs activity concentration in the Sea of Japan (SOJ) has been increasing since 2012/2013 and reached a maximum in 2015/2016 of approximately 3 Bq m$^{-3}$, which is above the pre-Fukushima accident level of the $^{137}$Cs activity concentration, 1.5 Bq m$^{-3}$. The $^{134}$Cs/$^{137}$Cs activity ratios that ranged from 0.36 to 0.51 in the ECS and the SOJ are evidence that the increase in the $^{137}$Cs activity concentration was derived from the Fukushima accident.

The increase in the FNPP1-$^{137}$Cs activity concentrations was first observed in the subsurface layer with a density of approximately around 25.2 kg m$^{-3}$, corresponding to the STMW, in the southern part of the Kuroshio Current in the NPSJ in 2012-2013, and decreases in the $^{137}$Cs activity concentrations in the NPSJ have been occurring since 2014. In the ECS, an increase in FNPP1-$^{137}$Cs was observed in 2014, and decreases in the $^{137}$Cs activity concentrations in the ECS have been observed since 2016. Note that the vertical distributions of the $^{137}$Cs activity concentrations in the ECS were almost constant or contained a subsurface peak. These results indicated that there was a 1-2 year time lag between when the high $^{137}$Cs activity concentrations occurred in the NPSJ and ECS. The increase in the $^{137}$Cs activity concentrations in the SOJ began in

2012 and continued until 2016. A time lag in the propagation of FNPP1-[137]Cs of approximately one year was observed in the SOJ. The similar [134]Cs/[137]Cs activity ratios in the surface layers of the ECS, SOJ, and Ogasawara region support an identical source of radiocaesium in these regions.

The total amount of recirculated FNPP1-[137]Cs from 2012 to 2016 was estimated to be 0.21±0.01 PBq, which
corresponds to 5.0 % of the FNPP1-[137]Cs in the STMW. Of this, the amounts of FNPP1-[137]Cs transported to the west and east along the TWC were almost the same or slightly larger along the eastern TWC. Of the FNPP1-[137]Cs transported into the SOJ, approximately 43 % (2.1 % of the total amount in STMW) passed through the Tsugaru Strait and back into the North Pacific Ocean. The FNPP1-[137]Cs transported northward was estimated to 14 % of the amount transported into the SOJ (0.7 % of the total amount in STMW).

The most important result of this study was the determination that FNPP1-[137]Cs was rapidly transported into the SOJ within several years after the FNPP1 accident.

**Acknowledgements**

The authors thank for collection of seawater samples by Miyuki Takahashi and Shun-pei Tomita at Oga Aquarium, Akita, Japan, staff at Kinosaki Aquarium, Hyogo, Japan, Hajime Chiba at Toyama Kosen, Toyama, Japan, Shigeo Takeda and the Captain and crew of Nagasaki-maru, Nagasaki Univ., Mitsuru Hayashi and the Captain and crew of Fukae-maru Kobe Univ., Yuki Nikaido and crew of Sado Kisen, Niigata, Japan, Akira Wada and the Captain and crew of Ferry CQP of Marix Line, Kagoshima, Japan, Kenichi Sasaki and the Captain and crew of Ushio-maru, Hokkaido Univ., Hakodate, Japan and Keiri
Imai and the Captain and crew of Oshoro-maru, Hokkaido Univ., Hakodate, Japan. We thank to Prof. E. Oka at the University of Tokyo and anonymous reviewer to valuable comments. We also thank to Prof. N. Hirose at Kyusyu University to provide us seawater transport volume data. We also thank to Rika Hozumi for her work to extract radiocaesium from seawater samples. This research was financially supported by Grant-in-Aid for Scientific Research on Innovative Areas, "Interdisciplinary study on environmental transfer of radionuclides from the Fukushima Dai-ichi NPP Accident" (Project No.
25110511) of the Japanese Ministry of Education, Culture, Sports, Science and Technology (MEXT). This research was also supported by the cooperation program of Institute of Nature and Environmental Technology, Kanazawa University (JFY2016, 2017) and the cooperation program by Institute of Radiation Emergency Medicine, Hirosaki University (JFY2016, 2017).

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

Figure 1: Location of the sampling points after the FNPP1 accident. Large black circles are sites monitored by the Japanese government. Blue circles are sites monitored by the Korean government. Black triangles are sites with measured vertical profiles. Circle colors corresponds to the $^{137}$Cs activity concentrations measured in 2011. Open circles are sites measured after the FNPP1-accident. The area around Japan was divided into 3 regions: the SOJ, ECS, and NPSJ (<141.5∘E). The locations of monitoring stations are also plotted in the upper Figure.

Figure 2: Temporal variations in the $^{137}$Cs activity concentrations in surface seawater in the Sea of Japan during the period from1960 to 2016. Circles indicate measured values, and red circles indicate data measured after the accident (2011 March 11). Black circles indicate the 0.5-year average value. Standard deviations of data were removed to clearly show the temporal variation.

Figure 3: Temporal variations in the FNPP1-$^{137}$Cs activity in the surface seawater at the monitoring sites in Japan after 2011. FNPP1-$^{137}$Cs: (a) Tomari, Aomori, (b) Niigata, Ishikawa, Fukui, Shimane, (c) Saga, Kagoshima, 314-01, and (d) 105-11.

Figure 4: Temporal variations in the $^{134}$Cs/$^{137}$Cs activity concentration ratio in the surface seawater at the monitoring stations in Japan from Jan 2016 to Jun 2016. (a) Tomari, Aomori, (b) Niigata, Ishikawa, Fukui, Shimane, and (c) Saga, Kagoshima.

Figure 5: Temporal variations in the $^{137}$Cs activity concentrations at stations (a) 314-01 and (b) 105-11.

Figure 6: Vertical distributions of the $^{137}$Cs activity concentrations over the (a) depth profile in the NPSJ, (b) potential density profile in the NPSJ, (c) depth profile in the ECS, (d) potential density profile in the ECS, (e) depth profile at 105-01 along the WTWC, and (f) potential density profile at 105-01 along the western TWC. Colour indicates the collection time (year).

Figure 7: Hovmoller diagrams of the $^{137}$Cs activity concentrations at a potential density along with an eastern TWC at a potential density of 25.2 ±0.5 kg m$^{-3}$. The ECS stations described on the x-axis are Kagoshima and Saga stations. Southwestern SOJ includes the monitoring stations Shimane, Fukui, Ishikawa, and Niigata. The northwestern SOJ includes the monitoring stations Aomori and Tomari. Color indicates the $^{137}$Cs activity concentrations (Bq m$^{-3}$).

Figure 8: Schematic diagram of the FNPP1-$^{137}$Cs transport in the North Pacific Ocean. The bold line indicates the Kuroshio pathway. The thin lines indicate the flow pathway of the Tsushima Warm Current. The circle with red dotted line are STMW formation area. The dotted green circle indicates the CMW formation area. Unfilled blue arrows indicate the FNPP1-$^{137}$Cs transport route deduced in this study. The red circles indicate the entrance region in the ECS. The estimated accumulated

flux during the period from 2012 to 2016 at each section is shown in the parenthesis. The inventory in the STMW (*) was deduced by Kaeriyama et al. (2016).

Figure 9: Latitudinal and horizontal distributions of the $^{134}$Cs/$^{137}$Cs activity ratios measured at the coastal sites of the SOJ and ECS in 2015-2016. The values were radioactive decay corrected to 11$^{th}$ March, 2011. The data measured in the Ogasawara area (red circles in (b)) were also added. (a) latitudinal distribution, (b) horizontal distribution.

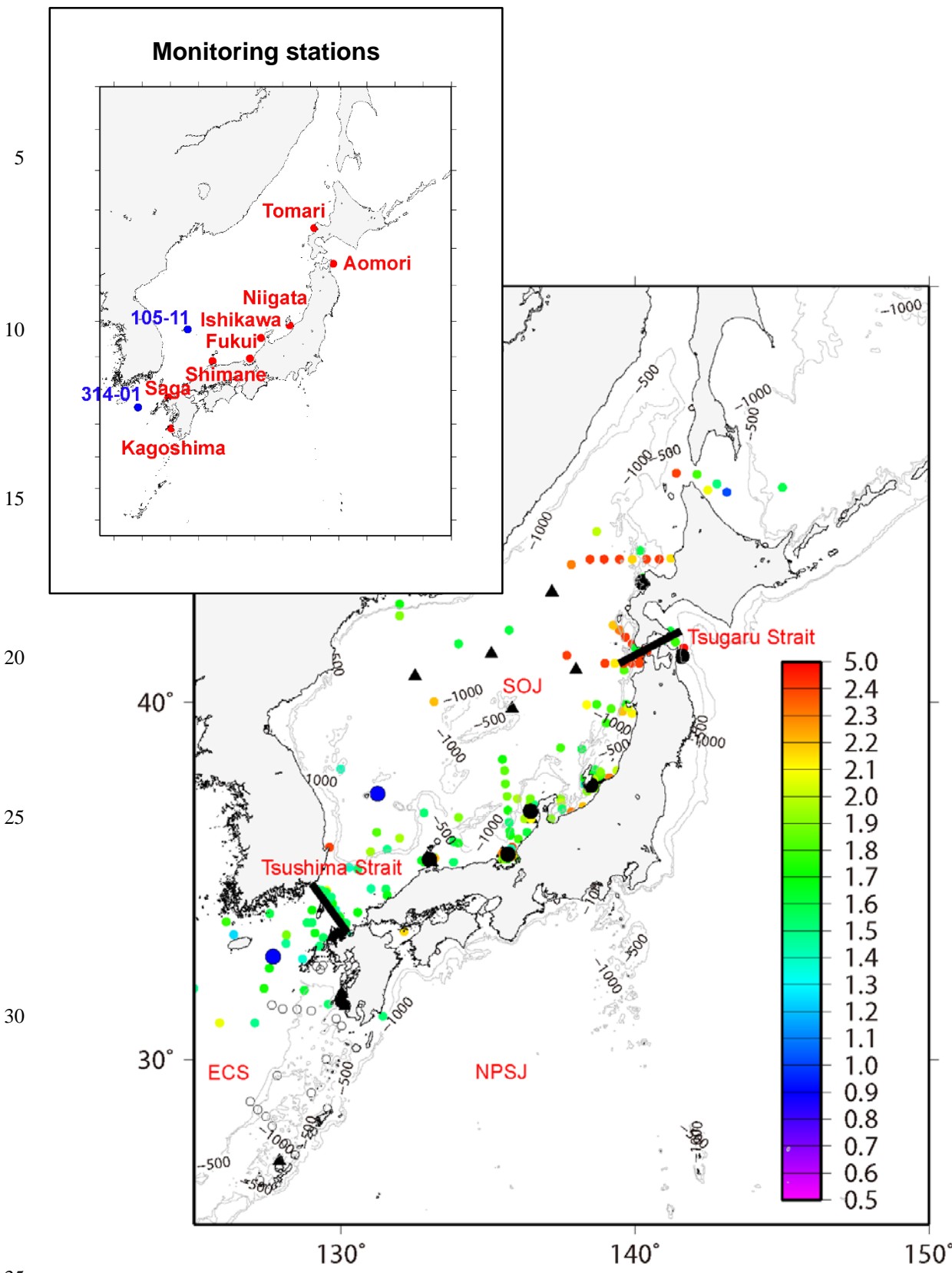

Figure 1: Location of the sampling points after the FNPP1 accident. Large black circles are sites monitored by the Japanese government. Blue circles are sites monitored by the Korean government. Black triangles are sites with measured vertical profiles. Circle colours correspond to the $^{137}$Cs activity concentrations measured in 2011. Open circles are sites measured after the FNPP1 accident. The area around Japan was divided into 3 regions: the SOJ, ECS, and NPSJ (<141.5°E). The locations of monitoring stations are also plotted in the upper Figure.

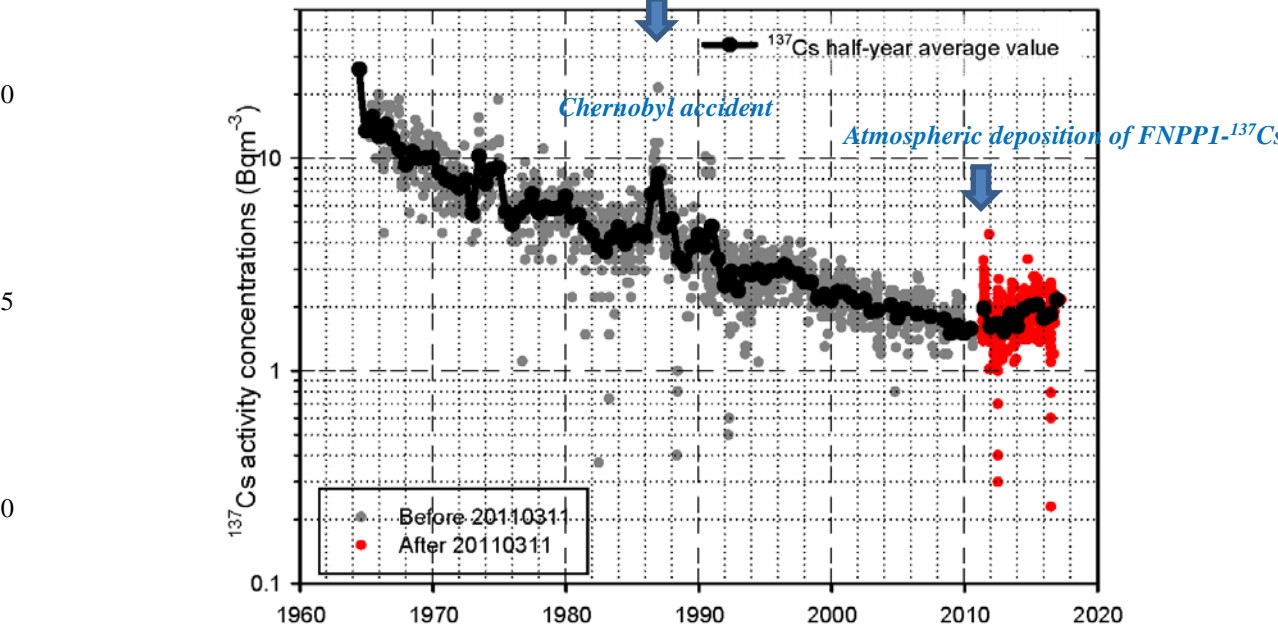

Figure 2: Temporal variations in the $^{137}$Cs activity concentrations in surface seawater in the Sea of Japan during the period from 1960 to 2016. Circles indicate measured values, and red circles indicate data measured after the accident (2011 March 11). Black circles indicate the 0.5-year average value. Standard deviations of data were removed to clearly show the temporal variation.

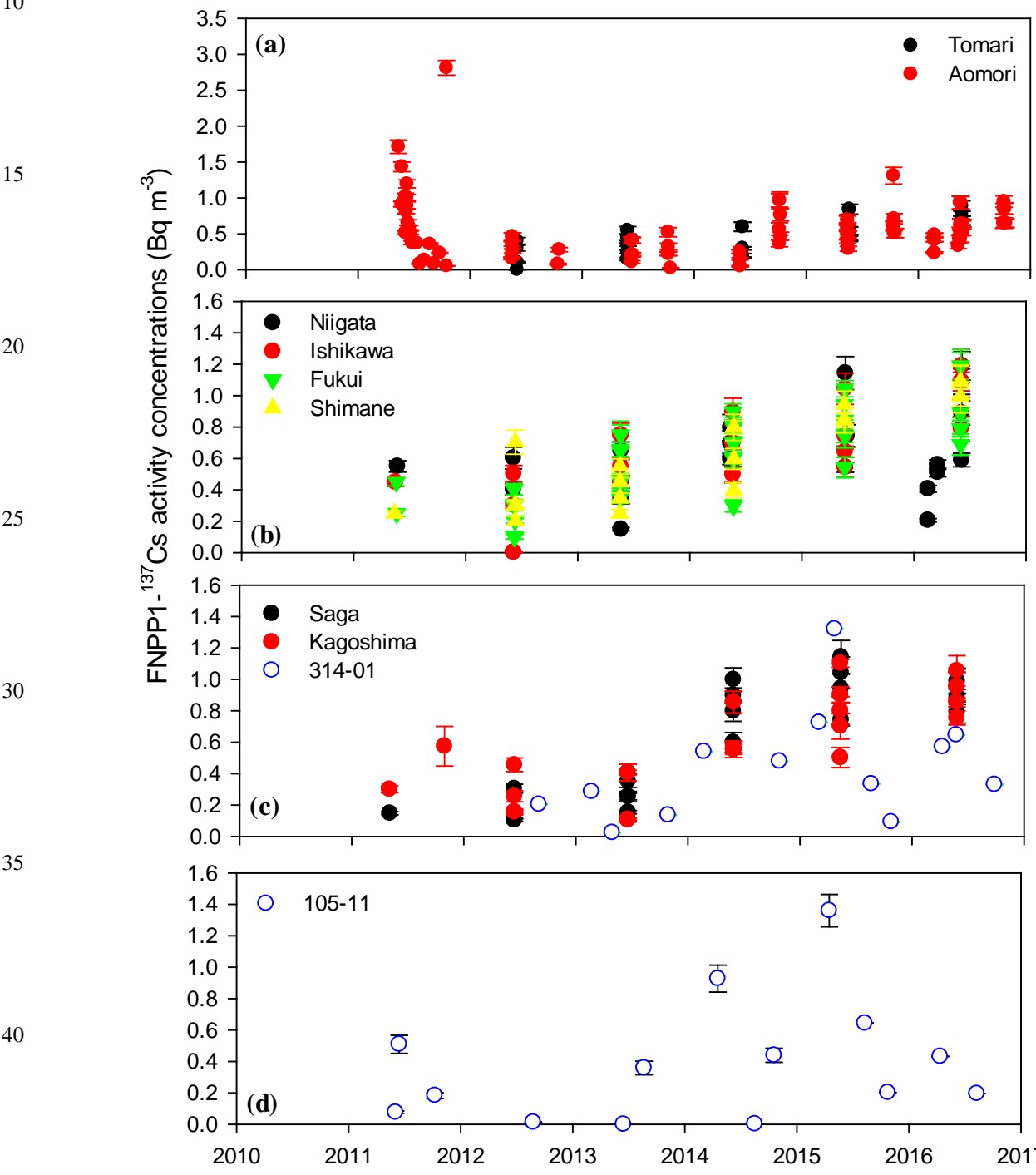

Figure 3: Temporal variations in the FNPP1-$^{137}$Cs activity concentrations in the surface seawater at the monitoring stations in the SOJ after 2011. (a) Tomari, Aomori, (b) Niigata, Ishikawa, Fukui, Shimane, (c) Saga, Kagoshima, 314-01, and (d) 105-11.

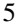

Figure 4: Temporal variations in the $^{134}$Cs/$^{137}$Cs activity concentration ratio in the surface seawater at the monitoring sites in Japan from Jan 2016 to Jun 2016: (a) Tomari, Aomori, (b) Niigata, Ishikawa, Fukui, Shimane, and (c) Saga, Kagoshima.

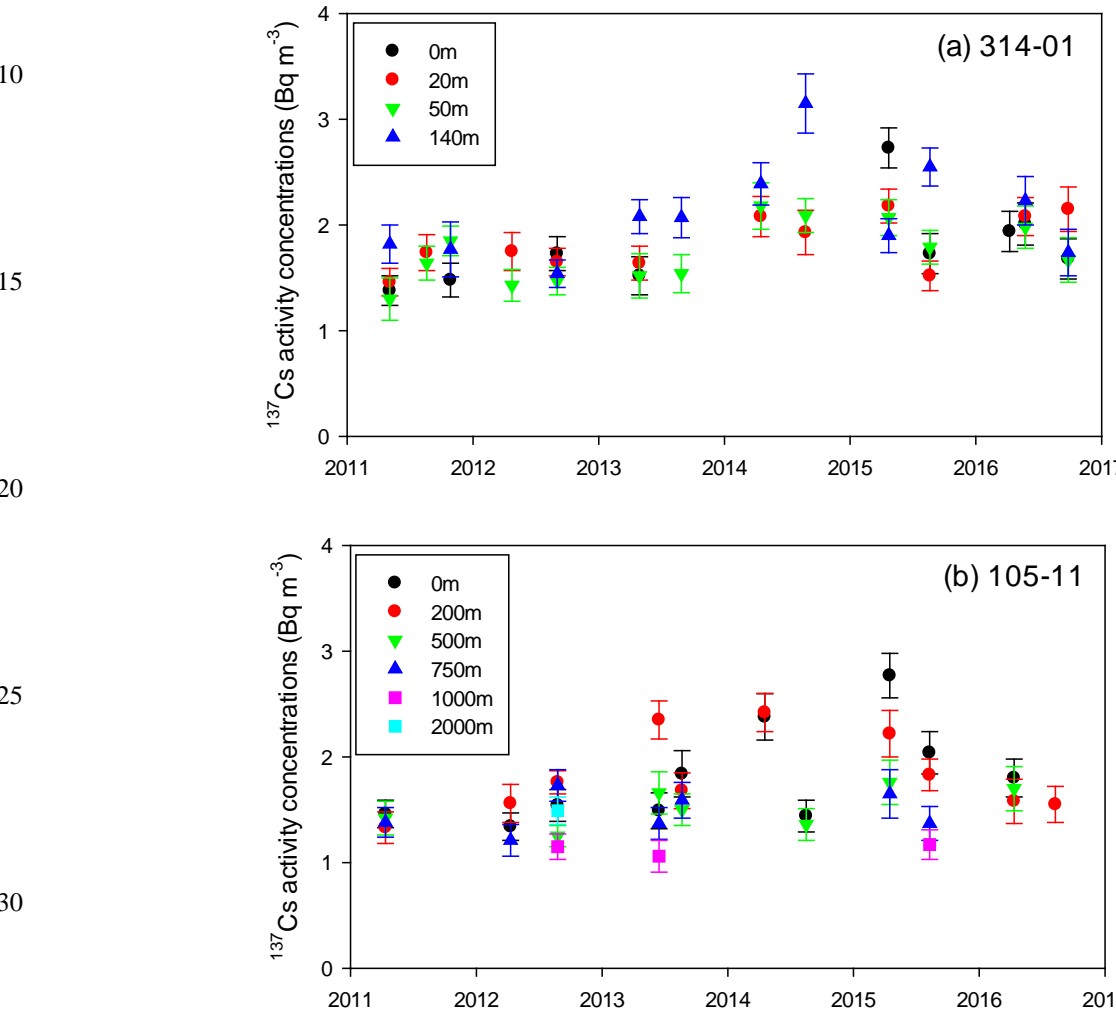

Figure 5: Temporal variations in the $^{137}$Cs activity concentrations at sites (a) 314-01 and (b) 105-11.

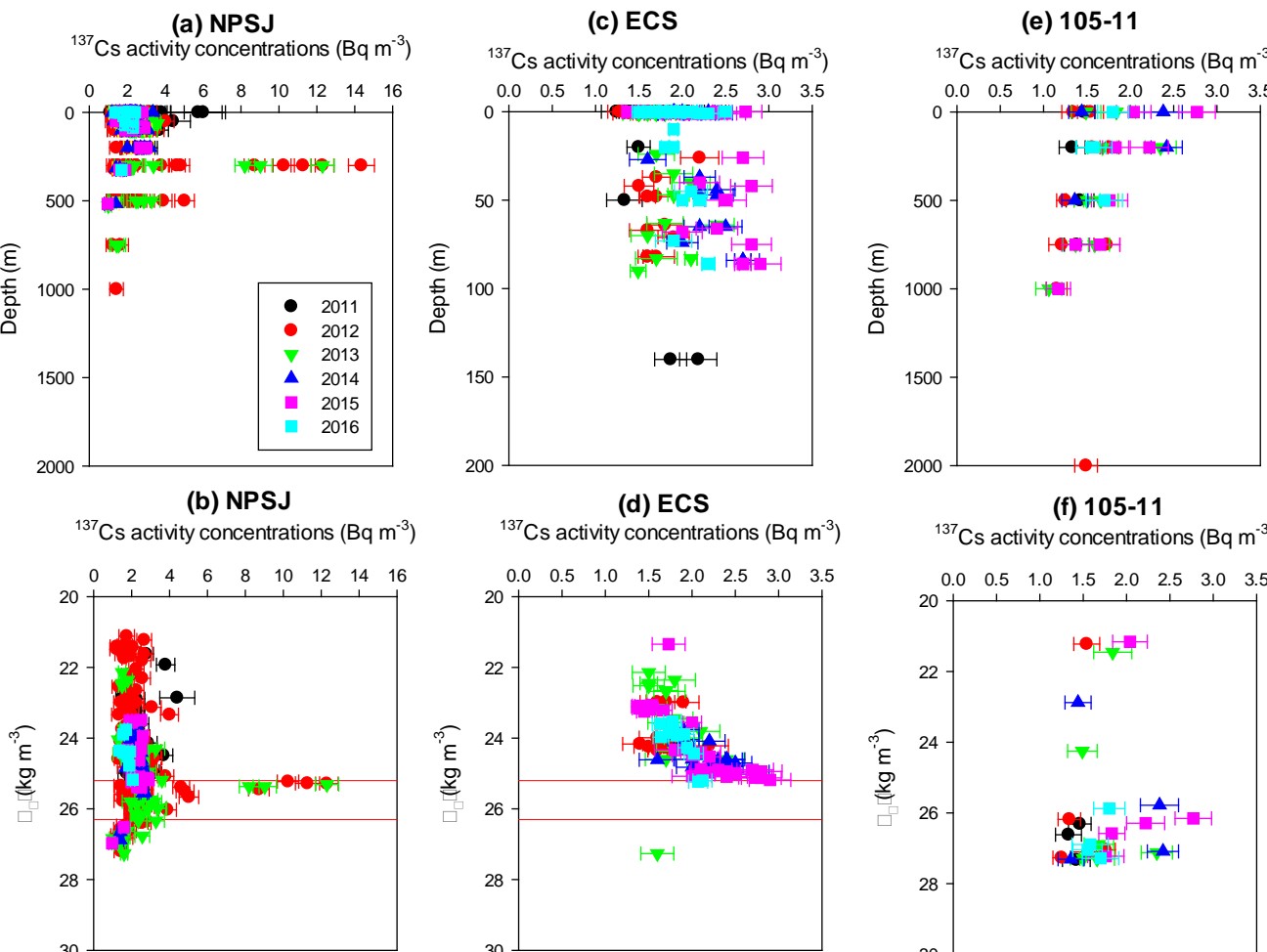

Figure 6: Vertical distributions of the $^{137}$Cs activity concentrations over the (a) depth profile in the NPSJ, (b) potential density profile in the NPSJ, (c) depth profile in the ECS, (d) potential density profile in the ECS, (e) depth profile at 105-01 along the WTWC, and (f) potential density profile at 105-01 along the western TWC. Colour indicates the collection time (year). Horizontal red line shown in (b) and (d) means the boundary value of $\sigma_\theta$ of the STMW.

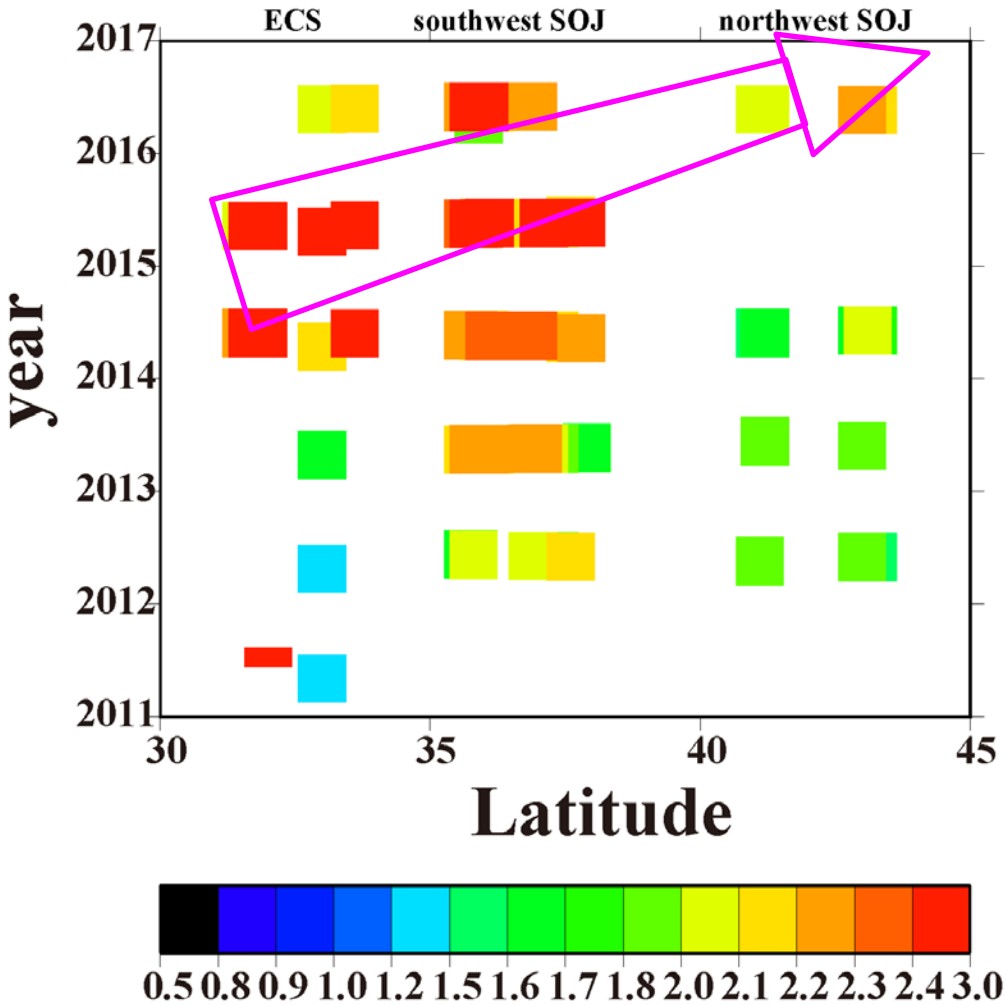

Figure 7: Hovmoller diagram of the [137]Cs activity concentrations at a potential density along with an eastern TWC at a potential density of 25.2 ±0.5 kg m[-3]. The ECS stations described on the x-axis are Kagoshima and Saga stations. Southwestern SOJ includes the monitoring stations Shimane, Fukui, Ishikawa, and Niigata. The northeastern SOJ includes the monitoring stations Aomori and Tomari. Color indicates the [137]Cs activity concentrations (Bq m[-3]). Arrow indicate the propagation of FNPP1-[137]Cs transport from ECS to SOJ.

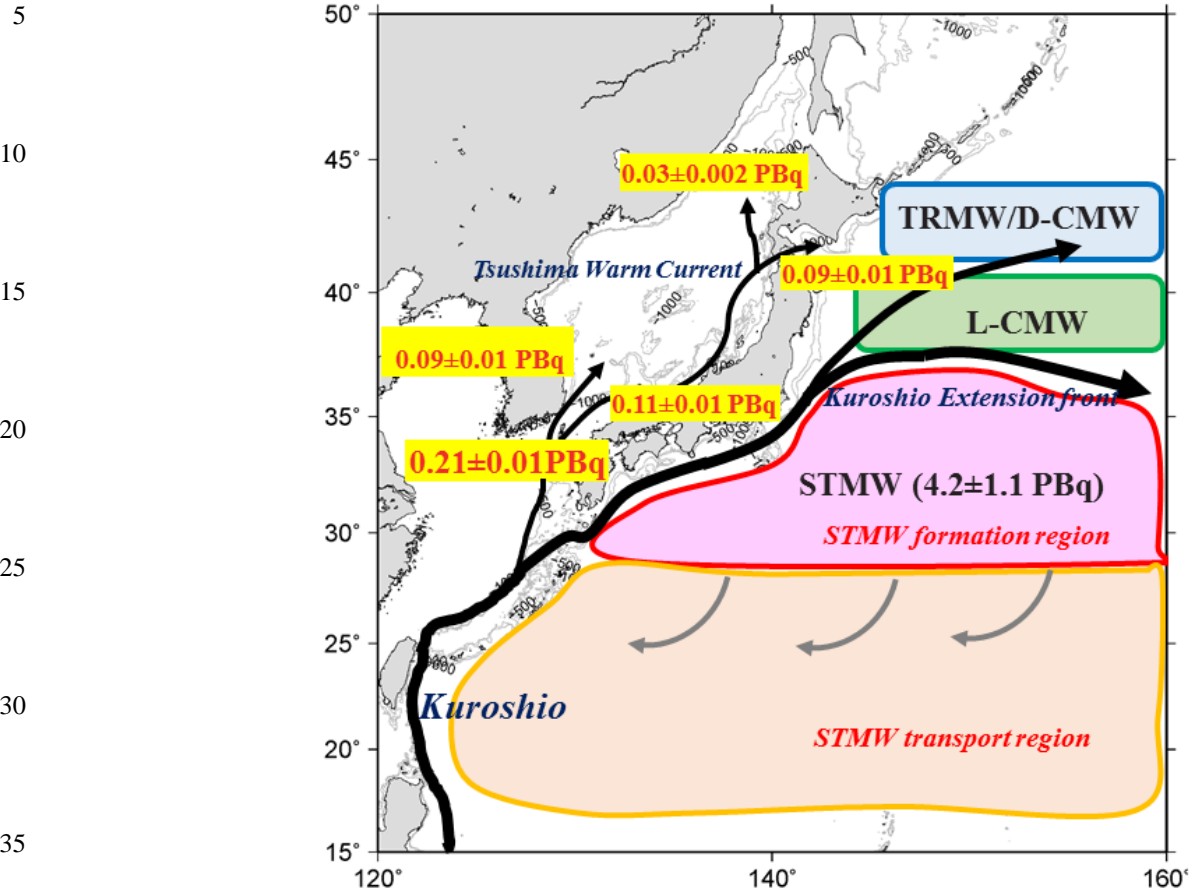

Figure 8: Schematic diagram of the FNPP1-[137]Cs transport in the North Pacific Ocean. The bold line indicates the Kuroshio pathway. The thin black lines indicate the flow pathway of the Tsushima Warm Current. The pink region is STMW formation area. The green region is the L-CMW formation area, and the blue region indicates the TRMW and D-CMW formation area. The estimated accumulated flux during the period from 2012 to 2016 at each section is shown. The inventory in the STMW was deduced by Kaeriyama et al. (2016).

(a)

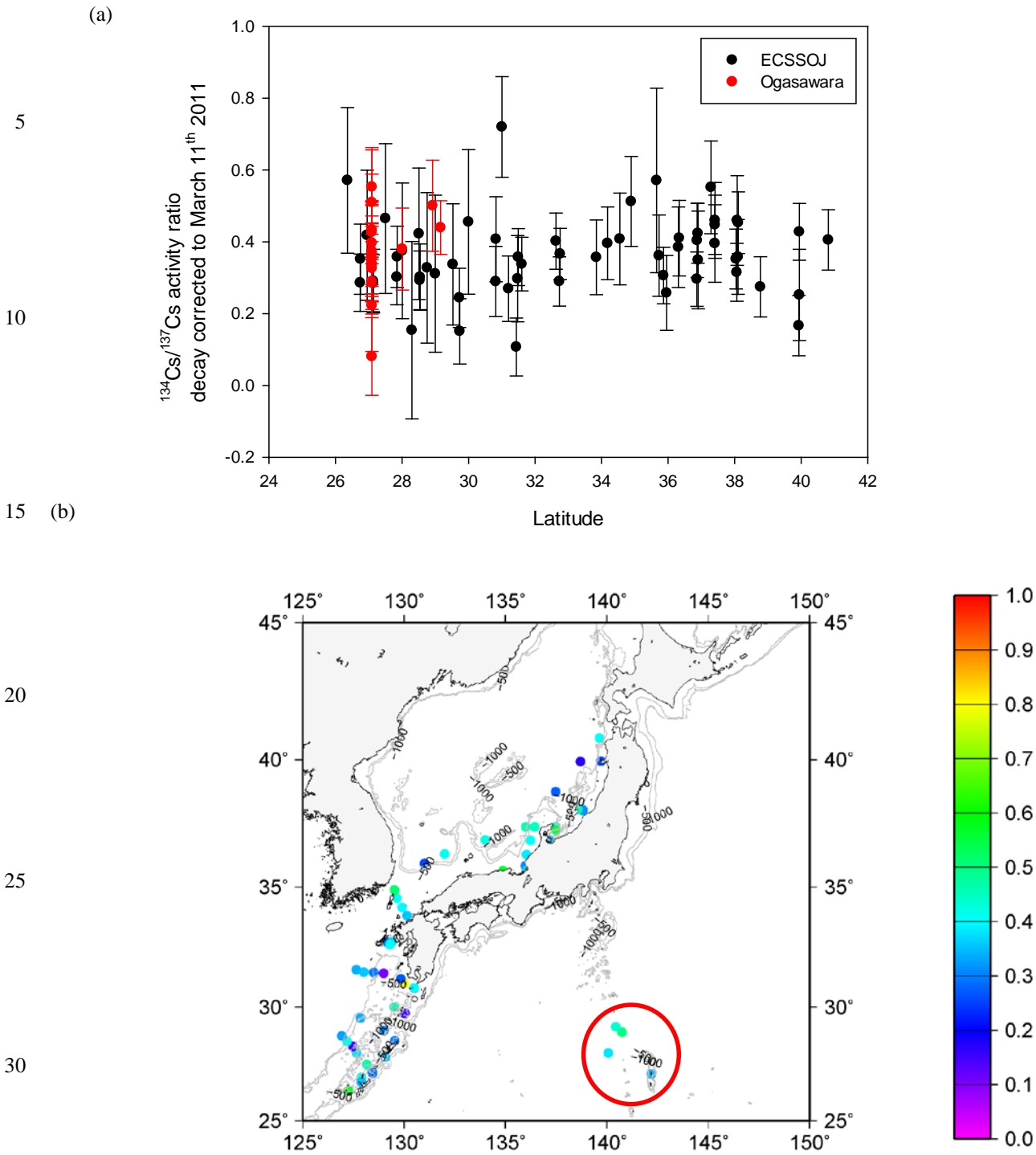

(b)

Figure 9: Latitudinal and horizontal distributions of the $^{134}$Cs/$^{137}$Cs activity ratios measured at the coastal sites of the SOJ and ECS in 2015-2016. The values were radioactive decay corrected to 11$^{th}$ March 2011. The data measured in the Ogasawara area (red circles in (a)) were also added. (a) Latitudinal distribution, (b) horizontal distribution.

Table 1. Estimated transport amount of FNPP1-$^{137}$Cs in the monitoring site in the SOJ along with the Tsushima Warm Current.

| Flow | Station | Transport amount (PBq)* | Ratio against to STMW(%) | Ratio against to inflow into the SOJ (%) |
|---|---|---|---|---|
| | | *Inflow to SOJ* | | |
| ECS to SOJ | Saga/304-01 | 0.21 ± 0.01 | 5.0 ± 2.3 | |
| | | *At Tsushima Strait* | | |
| Western TWC | 105-11 | 0.09 ± 0.01 | 2.1 ± 1.2 | 43 ± 10 |
| Eastern TWC | Shimane | 0.11 ± 0.01 | 2.6 ± 1.4 | 50 ± 7 |
| | | *Two branches of eastern TWC* | | |
| Outflow to Pacific Oceans | Aomori off | 0.09 ± 0.01 | 2.1 ± 1.2 | 43 ± 10 |
| Northward Transport | Tomari | 0.03 ± 0.002 | 0.7 ± 0.3 | 14 ± 2 |

*The value was decay corrected to 11th March 2011. ECS were estimated by using the average value of 314-01 and Saga monitoring sites. The transport amount (*) were estimated by sum of the duration from 2012 to 2016.