# Peer review of "Rapid transport of FNPP1-derived radiocaesium from subtropical mode water in the western North Pacific Ocean to the Sea of Japan"

_Ocean Science, 2017_

## Referee Comment (RC1) · E. Oka (Referee) · 30 Dec 2017

In this paper, the authors analyze radiocaesium, which originated mainly from nuclear bomb tests during 1950's-1960's and the Fukushima Nuclear Power Plant Accident in March 2011, in seas around Japan, specifically in the North Pacific south and east of Japan, the East China Sea, and the Japan Sea. The main finding of the manuscript, inferred from the title, seems to be the transport of subtropical mode water (STMW) from the North Pacific to the East China Sea through the Tokara Strait between Kyushu and the Ryukyu Islands and further to the Japan Sea, but I do not think that this finding

is fully supported by the data.

This finding is probably based on Fig. 6 showing that the 137Cs activity concentration distribution in the East China Sea was relatively high north of 30N and low south of 30N. However, the Tokara Strait is the place where the Kuroshio rapidly flows eastward from the East China Sea to the North Pacific. If STMW transported westward by the Kuroshio recirculation approaches the strait, how can it cross the intense Kuroshio front? A more plausible explanation is that STMW containing radiocaesium reaches the western boundary at lower latitudes, is entrained into the Kuroshio, and is transported to the west of Kyushu by the Tsushima Warm Current bifurcated from the Kuroshio.

Other major deficits of the manuscript are:

1. There are too many grammar errors and typos throughout the manuscript.

2. I do not well understand what the words "recirculation" and "timescale" in this manuscript mean. In physical oceanography, recirculation means the return flow associated with an intense western boundary current, but the word seems to be used for "transportation" in this manuscript. Also, timescale usually means a cycle of temporal variability, but this manuscript does not treat any variability. All "timescale" in this manuscript can be deleted. For example, it is more appropriate to rewrite the first sentence of the abstract to "137Cs derived from the Fukushima Nuclear Power Plant Accident rapidly transported to the Sea of Japan several years after its release to the environment in March 2011."

3. In Sec. 1, the authors need to explain which part of the Japan Sea was influenced by atmospheric deposition in March 2011. They also need to explain clearly what the main purpose of this study, with necessary explanation of its background.

4. Section 2 does not explain at what depths radiocaesium was sampled, especially at the coastal monitoring stations maintained by the Japanese and Korean governments (Fig. 7). In addition, readers cannot understand in which part of the East China Sea

(especially south of 30N) radiocaesium was sampled from Fig. 6, which does not show islands or bathymetry. These make it very difficult to understand the analysis in Sec. 3.2.

5. Although the aim of this paper is important to understand the "fate" of STMW (e.g., Gary et al., 2014, Journal of Physical Oceanography), the authors should understand that STMW is characterized by its low stratification or low potential vorticity, and not all water with potential density of 25.0-25.6 kg m-3 is STMW. When STMW is entrained into the Kuroshio at the western boundary, it rapidly loses its low potential vorticity and is not STMW anymore.

---

## Referee Comment (RC2) · Anonymous Referee #2 · 17 Jan 2018

**Review of the manuscript OS-2017-90 entitled "Rapid recirculation of FNPP1 derived radiocaesium suggesting new pathway of subtropical mode water in the western North Pacific to the Sea of Japan" by Inomata et al.**

The authors of the above manuscript examine the concentrations of radiocaesium as revealed by extensive sampling campaigns and monitoring programs around Japan (e.g. Pacific Ocean, East China Sea and Sea of Japan) to document transport pathways of mode waters and to estimate flow rates of radiocaesium through the neighboring straights (Tshushima and Tsugaru). Based on their impressive compiled dataset of decay-corrected radioactivity concentrations, and considering their summarized view of the most probable sources (e.g. the Fukushima Nuclear Power Plant accident in 2011), they conclude by presenting a simplified schematic of the circulation in the region and by proposing a new transport pathway for subtropical mode waters.

While the general topic of analyzing transport and mixing processes in the ocean based on almost-conservative tracers is very relevant, the manuscript is quite descriptive and I found that the results are not necessarily well supported by the analyses. In addition, many references are missing, thus the results are poorly-discussed results. The paper is not well written and contains too many grammar and typing mistakes as well as confusing sentences. Although the scientific question and the dataset are promising, this paper suffers from unsatisfactory writing, from superficial analyses and incomplete discussion. Both "result" and "discussion" sections should be developed by introducing new materials and extensive referencing. Overall, I think the manuscript is not publishable in OS in the present form. Before being acceptable, a major revision should carefully address the below points.

**Major points:**

- Although English is not my mother tongue, I found that grammar and typing mistakes are everywhere in the manuscript; furthermore, it contains many awkward sentences that confuse the reader and make the scientific message difficult to understand. I suggest the authors to consider English professional editing for the revised version.

- Analyses: to properly analyze the circulation and to estimate transport (by the way, how this is done in this manuscript?), especially through the straights or across frontal structure, I suggest the authors to present cross-sections (longitude or latitude versus depth) of radiocaesium. Looking at fig. 1, it seems that the authors have access to at least a few almost-synoptic measurements along "sections" (including at the straights), so this seems feasible. There are also many other ways to gain insights into transport processes in the ocean. For surface pathways, one could be looking at satellite data (SST patterns and SSH data to derive geostrophic currents). For subsurface pathways, one could examine the subsurface dispersal of ARGO floats in the region and exploit numerical models. The latter includes (1) exploring the outputs of Eulerian mesoscale models (looking at the list of references, some outputs may be sourced by the authors themselves; the Japanese modelling group at the Earth-Simulator possibly run routinely models in the region; there are also some outputs publicly available, for instance on http://marine.copernicus.eu/services-portfolio/access-to-products/) as well as (2) performing ad-hoc Lagrangian experiments to unveil pathways (see also some references provided below). Altogether, these additional analyses could really help to grasp the three-dimensional structure of the flow and could provide further evidences to support their conclusions.

- Discussion: I found that the results could be discussed further. For instance, the authors assumed

that atmospheric deposition occurred only over the Pacific but what if airborne radiocaesium also felt down over the Sea of Japan? They conclude with a transport pathway crossing a well-established oceanic front : what kind of physical processes (front destabilization? Mesoscale processes? Etc…) could explain this route? What about discussing the effect of diffusive mixing?... A recent publication (Sania et al. PNAS 2017) suggests that continuous submarine groundwater discharge could also contribute to the radioactive elements measured in Japanese coastal waters; this process could also occur on the western shores…etc...

– Bibliography: this contribution contains a large number of auto-citations and crucially suffers from a lack of key references. A large body of bibliography is not cited nor discussed properly. Bibliographic items which have been totally omitted in the present manuscript, but which MUST be cited and properly discussed in a revised, include:

Behrens, E.; Schwarzkopf, F. U.; Lubbecke, J. F.; Boning, C. W. Model simulations on the long-term dispersal of 137Cs released into the Pacific Ocean off Fukushima. Environ. Res. Lett. 2012, 7, 034004.

Budyansky, M.V., V.A. Goryachev, D.D. Kaplunenko, V.B. Lobanov, S.V. Prants, A.F. Sergeev, N.V. Shlyk, M.Yu. Uleysky, Role of mesoscale eddies in transport of Fukushima-derived cesium isotopes in the ocean, Deep Sea Research Part I: Oceanographic Research Papers, Volume 96, 2015, Pages 15-27, https://doi.org/10.1016/j.dsr.2014.09.007.

Oka, E., Qiu, B., 2012. Progress of North Pacific mode water research in the past decade. J. Oceanogr. 68, 5–20, http://dx.doi.org/10.1007/s10872-011-0032-5.

Oka, E., Qiu, B., Kouketsu, S., Uehara, K., Suga, T., 2012. Decadal seesaw of the Central and Subtropical Mode Water formation associated with the Kuroshio Extension variability. J. Oceanogr. 68, 355–360.

Prants, S.V., M.V. Budyansky, V.I. Ponomarev, M.Yu. Uleysky, P.A. Fayman. Lagrangian analysis of the vertical structure of eddies simulated in the Japan Basin of the Japan/East Sea. Ocean Modelling. V.86 pp.128-140 (2015). http://dx.doi.org/10.1016/j.ocemod.2014.12.010.

Prants, S.V., M.V. Budyansky, M.Yu. Uleysky. Statistical analysis of Lagrangian transport of subtropical waters in the Japan Sea based on AVISO altimetry data. Nonlin. Processes Geophys. V.24, p. 89-99, 2017 doi:10.5194/npg-24-1-2017.

Rossi, V.; Van Sebille, E.; Sen Gupta, A.; Garçon, V.; England, M. H. Multi-decadal projections of surface and interior pathways of the Fukushima Cesium-137 radioactive plume. Deep Sea Res., Part I 2013, 80, 37−46.

– Some figures are not necessarily well chosen and some are not very informative due to their poor content and/or low quality and low visual rendering. This is especially true for fig. 2 (e.g. grey and red data points and error bars are not readable); fig. 3 (panels d, e and f: cut the x-axis in mid-2016 since there is no data afterwards); fig. 5 (subplots are not numbered; what does black color means in panel 5c and 6c?); the scatters in fig. 9 are not clear to me, please clarify.

---

## Author Comment (AC1) · 8 Mar 2018

Thank you very much for your review to our manuscript. I was revised the manuscript according to your comment. The modified part are shown in this document.

1. There are too many grammar errors and typos throughout the manuscript. A. The manuscript was checked by English native speakers.

2. I do not well understand what the words "recirculation" and "timescale" in this manuscript mean. In physical oceanography, recirculation means the return flow as-

sociated with an intense western boundary current, but the word seems to be used for "transportation" in this manuscript. Also, timescale usually means a cycle of temporal variability, but this manuscript does not treat any variability. All "timescale" in this manuscript can be deleted. For example, it is more appropriate to rewrite the first sentence of the abstract to "137Cs derived from the Fukushima Nuclear Power Plant Accident rapidly transported to the Sea of Japan several years after its release to the environment in March 2011." A. The word "timescale" was deleted and the term "recirculation" was modified as "transportation".

3. In Sec. 1, the authors need to explain which part of the Japan Sea was influenced by atmospheric deposition in March 2011. They also need to explain clearly what the main purpose of this study, with necessary explanation of its background. A. We added 137Cs activity concentrations as well as the sampling location by usingcolor in Fig. 1. The colored circle represent the sample measured in 2011. (white open circles were measurement data after 2012; triangle is the point conducted to vertical profile; The monitoring sites by Japanese and Korean governments were also plotted. As shown in Fig. 1, higher 137Cs activity concentrations were observed in the north of 40oN.

[Page 16, Line 21] In the northeastern part of the SOJ, atmospherically deposited radiocaesium were caused to approximately 1-2 times higher activity concentrations of radiocaesium in May 2011 (1.5-2.8 Bqm-3) in northeastern SOJ than those of before the FNPP1 accident (∼1.5 Bqm-3) (Fig. 1). By the end of 2011, the 137Cs activity concentrations in the northeastern part of the SOJ had rapidly decreased to almost the same levels as those before the FNPP1 accident (Inoue et al., 2012). In fact, the 137Cs activity concentration in surface water at an SOJ coastal monitoring site after July 2011 was almost the same as the pre-FNPP1 accident level (http://radioactivity.nsr.go.jp/ja/list/195/list-8.html).

4. Section 2 does not explain at what depths radiocaesium was sampled, especially at the coastal monitoring stations maintained by the Japanese and Korean governments (Fig. 7). In addition, readers cannot understand in which part of the East China Sea.

(especially south of 30N) radiocaesium was sampled from Fig. 6, which does not show islands or bathymetry. These make it very difficult to understand the analysis in Sec. 3.2.

A. Seawater sampling methods are different with each organization. The monitoring data by the Japanese government (MARIS) was using bucket sampler. In Korea, seawater was sampled by Rosette sampler and/or intake sampling. The sampling by volunteer vessel was conducted by bucket or intake sampling. Therefore, we define the surface seawater as collecting less than 10 m. In order to show the sampling depth of seawater, the information of sampling depth was described in more detail. In order to clearly show the topographical information, the authors added the islands and bathymetry in the Map's Figure (such as Fig. 1, 8, 9, Fig.SI12)

Fig. 1 has already shown in Page 2.

[Page 3, Line 4-5] The term "surface seawater" used in this study defines a sample collected at less than 10 m depth. [Page 5, Line 1-4] Fig. 5 show the temporal variation in the 137Cs activity concentrations at different depths at station 314-01 in the ECS (Fig. 5a; 0-140 m) and station 105-11 in the SOJ (Fig. 5b; 0-2000 m). The increase in the 137Cs activity concentrations at the subsurface layer (140 m at the station 314-01, 200 m at the station 105-11) occurred 1 year earlier than that in the shallower layers.

5. Although the aim of this paper is important to understand the "fate" of STMW(e.g., Gary et al., 2014, Journal of Physical Oceanography), the authors should understand that STMW is characterized by its low stratification or low potential vorticity, and not all water with potential density of 25.0-25.6 kg m-3 is STMW. When STMW is entrained into the Kuroshio at the western boundary, it rapidly loses its low potential vorticity and is not STMW anymore. A. Thank you very much for the adequate comments. In SJPN, the peak of 137Cs activity concentrations was observed in the seawater corresponds to the potential density 25.2 kg m-3. Therefore, we can consider that 137Cs existed in the STMW. In ECS, 137Cs was mainly observed in the layer having similar potential

density. However, it was revealed that 137Cs in the SOJ was observed in denser layer. As the reviewer mentioned, these distributions pattern clearly show the FATE of STMW in the SOJ. Section 4 was largely modified.

[Page 5, Line 15-29] The vertical distributions of the 137Cs activity concentration over the depth and potential density (ïĄşïĄśïĂň kg m-3) profiles in the southern Japan (SJPN), in the ECS, and at the station 105-11 are shown in Fig. 6. In the SJPN, the 137Cs activity concentrations showed subsurface maxima at approximately 300 m depth in 2012-2013, with activities of 8.2-12.3 Bqm-3 as shown in Fig. 6a. These high 137Cs activity concentrations were measured in the region between 136-138°E and 26-30°N. After 2014, a subsurface peak in the 137Cs activity concentration was not observed. These subsurface peaks in the 137Cs activity concentrations were found in the layer corresponding to a potential density of 25.2 kg m-3 (Fig. 6b). This is consistent with previous findings of an 137Cs activity maximum in STMW with a potential density of 25.0-25.6 kg m-3 (Kaeriyama et al., 2014; Kumamoto et al., 2014). In the ECS, the 137Cs activity concentrations gradually increased beginning in 2012 and attained the maximum activity concentrations (2.9±0.24 Bqm-3) in 2015, following by a decreasing trend in the 137Cs activity concentrations in 2016, as shown in Fig. 6c. The higher 137Cs activity concentrations above 2 Bq m-3 in the ECS were found in the layer corresponding to a potential density of 23.6-25.2 kg m-3, as shown in Fig. 6d. On the other hand, the 137Cs activity concentrations at the station 105-11, located in the western SOJ, decreased with increasing depth until 500 m (Fig. 6e). The higher 137Cs activity concentrations at the station 105-11 were measured in 2014 and 2015, and the 137Cs activity concentrations decreased in 2016. The 137Cs activity concentrations above 2 Bq m-3 at the station 105-11, except one sample measured at 2015, were located in the layer with a potential density of 25.8-27.1 kg m-3 (Fig. 6f).

[Page 7, Line 22-35] As shown in Fig. 6e, the 137Cs activity concentrations at station 105-11, located along the western TWC in the SOJ, were maximum at the surface and gradually decreased with increasing depth. This vertical distribution

is different from those in the SJPN (Fig. 6a) and ECS (Fig. 6c). Particularly, the subsurface peak observed in the SJPN and ECS did not appear at station 105-11. At station 105-11, most of the 137Cs existed in seawater with a potential density of 25.7-27.3 kg m-3 (Fig. 6f), which was located in a higher potential density layer than that in the SJPN and ECS. A similar vertical distribution was also observed at the western coast of the Japanese Islands along the eastern TWC (Fig. SI11). These distributions were due to cooling in the surface layer after water was transported from the Tsushima Strait. Physical processes such as the convergence and subduction of surface water inside the eddies are important mechanisms of downward transport of radiocaesium (Miyao et al., 1998; Budyansky et al., 2015). Based on the lagrangian analysis of the vertical structure of the eddies in the SOJ, the eddy in summer was characterized as unstable layer results in thinner mixed layer depth and weaker seasonal pycnocline in upper layer. During winter season, the eddy became to be very stable in the upper layers, which leads to increase mixed layer depth and become to weak seasonal pycnocline (Prants et al., 2015). There is a possibility of the seasonality of downward transport of radiocaesium. As expected, the past global fallout-137Cs had already penetrated and accumulated in the deeper layers of the SOJ.

Please also note the supplement to this comment:
https://www.ocean-sci-discuss.net/os-2017-90/os-2017-90-AC1-supplement.pdf

**Supplement:**

**Replay to reviewer, Prof. Oka**

Thank you very much for your review to our manuscript. I was revised the manuscript according to your comment. The modified part are shown in this document.

**1. There are too many grammar errors and typos throughout the manuscript.**

A. The manuscript was checked by English native speakers.

**2. I do not well understand what the words "recirculation" and "timescale" in this manuscript mean. In physical oceanography, recirculation means the return flow associated with an intense western boundary current, but the word seems to be used for "transportation" in this manuscript. Also, timescale usually means a cycle of temporal variability, but this manuscript does not treat any variability. All "timescale" in this manuscript can be deleted. For example, it is more appropriate to rewrite the first sentence of the abstract to "137Cs derived from the Fukushima Nuclear Power Plant Accident rapidly transported to the Sea of Japan several years after its release to the environment in March 2011."**

A. The word "timescale" was deleted and the term "re-circulation" was modified as "transportation".

**3. In Sec. 1, the authors need to explain which part of the Japan Sea was influenced by atmospheric deposition in March 2011. They also need to explain clearly what the main purpose of this study, with necessary explanation of its background.**

A. We added $^{137}$Cs activity concentrations as well as the sampling location by usingcolor in Fig. 1. The colored circle represent the sample measured in 2011. (white open circles were measurement data after 2012; triangle is the point conducted to vertical profile; The monitoring sites by Japanese and Korean governments were also plotted. As shown in Fig. 1, higher $^{137}$Cs activity concentrations were observed in the north of 40ºN.

[Page 16, Line 21]

In the northeastern part of the SOJ, atmospherically deposited radiocaesium were caused to approximately 1-2 times higher activity concentrations of radiocaesium in May 2011 (1.5-2.8 Bqm$^{-3}$) in northeastern SOJ than those of before the FNPP1 accident (~1.5 Bqm$^{-3}$) (Fig. 1). By the end of 2011, the $^{137}$Cs activity concentrations in the northeastern part of the SOJ had rapidly decreased to almost the same levels as those before the FNPP1 accident (Inoue et al., 2012). In fact, the $^{137}$Cs activity concentration in surface water at an SOJ coastal monitoring site after July 2011 was almost the same as the pre-FNPP1 accident level (http://radioactivity.nsr.go.jp/ja/list/195/list-8.html).

[Figure]

Fig. 1.

[Figure]

Figure 1: Location of the sampling points after the FNPP1 accident. Large red circles are sites monitored by the Japanese government. Blue circles are sites monitored by the Korean government. Black triangles are sites with measured vertical profiles. Circle colors corresponds to the $^{137}$Cs activity concentrations measured in 2011. Open circles are sites measured after the FNPP1-accident. The area around Japan was divided into 3 regions: the SOJ, ECS, and SJPN (<141.5∘E).

4. **Section 2 does not explain at what depths radiocaesium was sampled, especially at the coastal monitoring stations maintained by the Japanese and Korean governments (Fig. 7). In addition, readers cannot understand in which part of the East China Sea. (especially south of 30N) radiocaesium was sampled from Fig. 6, which does not show islands or bathymetry. These make it very difficult to understand the analysis in Sec. 3.2.**

A. Seawater sampling methods are different with each organization. The monitoring data by the Japanese government (MARIS) was using bucket sampler. In Korea, seawater was sampled by Rosette sampler and/or intake sampling. The sampling by volunteer vessel was conducted by bucket or intake sampling. Therefore, we define the surface seawater as collecting less than 10 m. In order to show the sampling depth of seawater, the information of sampling depth was described in more detail. In order to clearly show the topographical information, the authors added the islands and bathymetry in the Map's Figure (such as Fig. 1, 8, 9, Fig.SI12)

Fig. 1 has already shown in Page 2.

[Figure]

Figure 9: Latitudinal and horizontal distributions of the $^{134}$Cs/$^{137}$Cs activity concentration ratios measured at the coastal sites of the SOJ and ECS in 2015-2016. The values were radioactive decay-corrected to 11$^{th}$ March, 2011. The data measured in the Ogasawara area (red circles in (a)) were also added. (a) Latitudinal distribution, (b) horizontal distribution.

[Figure]

Figure SI12. Horizontal distributions of $^{137}$Cs activity concentrations in SJPN and ECS from 2011 to 2016. Circles mean the surface data. Square denotes stations having vertical distribution. Unit is Bq m$^{-3}$. Higher activity concentrations around 25-30ºN and 135-140ºE were measured in the year of 2012.

[Page 3, Line 4-5]

The term "surface seawater" used in this study defines a sample collected at less than 10 m depth.

[Page 5, Line 1-4]

Fig. 5 show the temporal variation in the $^{137}$Cs activity concentrations at different depths at station 314-01 in the ECS (Fig. 5a; 0-140 m) and station 105-11 in the SOJ (Fig. 5b; 0-2000 m). The increase in the $^{137}$Cs activity concentrations at the subsurface layer (140 m at the station 314-01, 200 m at the station 105-11) occurred 1 year earlier than that in the shallower layers.

5. **Although the aim of this paper is important to understand the "fate" of**

**STMW(e.g., Gary et al., 2014, Journal of Physical Oceanography), the authors should understand that STMW is characterized by its low stratification or low potential vorticity, and not all water with potential density of 25.0-25.6 kg m-3 is STMW. When STMW is entrained into the Kuroshio at the western boundary, it rapidly loses its low potential vorticity and is not STMW anymore.**

A. Thank you very much for the adequate comments. In SJPN, the peak of $^{137}$Cs activity concentrations was observed in the seawater corresponds to the potential density 25.2 kg m$^{-3}$. Therefore, we can consider that $^{137}$Cs existed in the STMW. In ECS, $^{137}$Cs was mainly observed in the layer having similar potential density. However, it was revealed that $^{137}$Cs in the SOJ was observed in denser layer. As the reviewer mentioned, these distributions pattern clearly show the FATE of STMW in the SOJ. Section 4 was largely modified.

[Page 5, Line 15-29]
The vertical distributions of the $^{137}$Cs activity concentration over the depth and potential density ($\sigma_{\theta}$, kg m$^{-3}$) profiles in the southern Japan (SJPN), in the ECS, and at the station 105-11 are shown in Fig. 6. In the SJPN, the $^{137}$Cs activity concentrations showed subsurface maxima at approximately 300 m depth in 2012-2013, with activities of 8.2-12.3 Bqm$^{-3}$ as shown in Fig. 6a. These high $^{137}$Cs activity concentrations were measured in the region between 136-138°E and 26-30°N. After 2014, a subsurface peak in the $^{137}$Cs activity concentration was not observed. These subsurface peaks in the $^{137}$Cs activity concentrations were found in the layer corresponding to a potential density of 25.2 kg m$^{-3}$ (Fig. 6b). This is consistent with previous findings of an $^{137}$Cs activity maximum in STMW with a potential density of 25.0-25.6 kg m$^{-3}$ (Kaeriyama et al., 2014; Kumamoto et al., 2014). In the ECS, the $^{137}$Cs activity concentrations gradually increased beginning in 2012 and attained the maximum activity concentrations (2.9±0.24 Bqm$^{-3}$) in 2015, following by a decreasing trend in the $^{137}$Cs activity concentrations in 2016, as shown in Fig. 6c. The higher $^{137}$Cs activity concentrations above 2 Bq m$^{-3}$ in the ECS were found in the layer corresponding to a potential density of 23.6-25.2 kg m$^{-3}$, as shown in Fig. 6d. On the other hand, the $^{137}$Cs activity concentrations at the station 105-11, located in the western SOJ, decreased with increasing depth until 500 m (Fig. 6e). The higher $^{137}$Cs activity concentrations at the station 105-11 were measured in 2014 and 2015, and the $^{137}$Cs activity concentrations decreased in 2016. The $^{137}$Cs activity concentrations above 2 Bq m$^{-3}$ at the station 105-11, except one sample measured at 2015, were located in the layer with a potential density of 25.8-27.1 kg m$^{-3}$ (Fig. 6f).

[Page 7, Line 22-35]
As shown in Fig. 6e, the $^{137}$Cs activity concentrations at station 105-11, located along the western TWC in the SOJ, were maximum at the surface and gradually decreased with increasing depth. This vertical distribution is different from those in the SJPN (Fig. 6a) and ECS (Fig. 6c). Particularly, the subsurface peak observed in the SJPN and ECS did not appear at station 105-11. At station 105-11, most of the $^{137}$Cs existed in seawater with a potential density of 25.7-27.3 kg m$^{-3}$ (Fig. 6f), which was located in a higher potential density layer than that in the SJPN and ECS. A similar vertical distribution was also observed at the western coast of the Japanese Islands along the eastern TWC (Fig. SI11). These distributions were due to cooling in the surface layer after water was transported from the Tsushima Strait. Physical processes such as the convergence and subduction of surface water inside the eddies are important mechanisms of downward transport of radiocaesium (Miyao et al., 1998; Budyansky et al., 2015). Based on the lagrangian analysis of the vertical structure of the eddies in the SOJ, the eddy in summer was characterized as unstable layer results in thinner mixed layer depth and weaker seasonal pycnocline in upper layer. During winter season, the eddy became to be very stable in the upper layers, which leads to increase mixed layer depth and become to weak seasonal pycnocline (Prants et al., 2015). There is a possibility of the seasonality of downward transport of radiocaesium. As expected, the past global fallout-$^{137}$Cs had already penetrated and accumulated in the deeper layers of the SOJ.

---

## Author Comment (AC2) · 8 Mar 2018

Thank you very much for valuable your comments and questions. According to your comments, the manuscript was revised. The answer to your comments are described as below.

Major points: – Although English is not my mother tongue, I found that grammar and typing mistakes are everywhere in the manuscript; furthermore, it contains many awkward sentences that confuse the reader and make the scientific message difficult to

understand. I suggest the authors to consider English professional editing for the revised version. A. The author ask to the English editing company to correct English.

– Analyses: to properly analyze the circulation and to estimate transport (by the way, how this is done in this manuscript?), especially through the straights or across frontal structure, I suggest the authors to present cross-sections (longitude or latitude versus depth) of radiocaesium.

A. In order to clearly show the propagation of FNPP1-137Cs, latitude-time cross sections at the potensial density surface were depicted in Figure 7. We selected the 25.20.5 kg m-3 surface data becasue the maximum 137Cs activity concentration was obsevred at this layer. In other wods, we think that the center of FNPP1-137Cs transported in the North Pacific Ocean was located in this layer. In this latitude-time cross sections, the 137Cs activity concentrations in the ECS were maximum in 2014/2015, and then tended to decrease in 2016. In southwest SOJ, 137Cs activity concentrations were gradually increased until 2016. In the northeastern Japanese monitoring stations, the 137Cs activity concentratins were gradualy increased. These suggest the propagation of the 137Cs ocurred form suthwestern SOJ and northeastern SOJ.

(Page 5, Line 30 ; Page 6, Line 5) Fig. 7 displays the latitude-time cross sections of the 137Cs activity concentrations at potential temperatures of 25.2±0.5 kgm-3 along the western TWC, 25.7±0.5 kgm-3 along the eastern TWC, and 26.7±0.5 kgm-3 along the eastern TWC. These potential density surfaces were selected to show the maximum 137Cs activity concentrations observed in the ECS. The vertical distributions of the 137Cs activity concentrations with depth and potential density at each monitoring station are displayed in Fig. SI11. Note that in the SOJ, the vertical distributions of the 137Cs activity concentrations below 250 m were almost constant, and the subsurface peak of 137Cs was not found at the monitoring stations along the eastern TWC (Fig. SI11). It is noted that the 137Cs activity concentrations before the FNPP1 accident were approximately 1.5 Bq m-3. As shown in Fig. 7a, in the ECS, the 137Cs activity concentrations gradually increased and attained the maximum in 2014/2015. The 137Cs activity concentrations in the ECS tended to decrease in 2016 in a layer with a density of 25.2±0.5 kgm-3. In the southwestern part of the SOJ (Shimane, Fukui, Ishikawa, Niigata), the 137Cs activity concentrations also gradually increased beginning in 2012, and the activity concentrations attained a maximum of 2.5 Bq m-3 in 2015/2016; these trends were almost the same as those at the monitoring stations in the ECS. In the northwestern SOJ (Aomori and Tomari), the 137Cs activity concentrations were slightly increased and higher activity concentrations above 2 Bq m-3 were observed in 2016. These results revealed that the propagation of FNPP1-137Cs occurred within several years from the ECS to the SOJ along the TWC. On the other hand, the latitude-time cross section along the western TWC indicated that the higher activity concentrations were observed in the layers with potential densities of 25.7±0.5 and 26.7±0.5 kgm-3 than other layers (Fig. 7b,c). Higher 137Cs activity concentrations were observed in the higher potential density layer in comparison with those in the seawater along with eastern TWC. In the 105-11 station, the decrease of 137Cs activity concentrations were started from 2015/2016.

(Page 7, Line 13-21) In this study, we revealed that FNPP1-137Cs entered the SOJ via the ECS; then, FNPP1-137Cs was transported northward with the TWC. In the SOJ, a time lag of the propagation of FNPP1-radiocaesium of approximately one year was observed (Fig. 7). Based on measurements of phosphate, one of the dominant seawater nutrients, Kodama et al. (2016) revealed that the phosphate concentrations in surface seawater during winter were significantly positively correlated with the concentrations in the saline ECS seawater in the preceding summer, and the surface water of the southern SOJ was almost entirely replaced by the ECS seawater during May–October. Kodama et al. (2016) suggested that the transport of water-soluble constituents from the ECS to the SOJ takes at least approximately 0.5 years. The propagation of FNPP1-radiocaesium in the SOJ was consistent with the propagation time scale of nutrients concentration change from the ECS to the SOJ (Kodama et al., 2016).

- Looking at Fig. 1, it seems that the authors have access to at least a few almost-synoptic measurements along "sections" (including at the straights), so this seems feasible.

A. The data used in this study was collected by literarture reserach except for our own data to measure the 134Cs activity concentrations. The most of sections data did not measure reapeatedly. Instead of this, we investigated the time vatiation at the monitoring sites by the Japanese and Korean government.

(Page 2, Line 41- Page 3, Line 16) 2.1ïïjŐData sources After the FNPP1 accident, many radiocaesium measurements were taken in the SOJ and western North Pacific Ocean (Fig. 1). To elucidate the temporal and spatial distributions of the radiocaesium activity concentrations, we use as many data points as possible. We, therefore, compiled all available data from the literature and reported studies. Most of the data before the FNPP1 accident was included in the database, "Historical Artificial Radionuclides in the Pacific Ocean and its Marginal Seas (HAM database)" (Aoyama and Hirose, 2003 and their updated version). The data observed after the FNPP1 accident were shown in Aoyama et al. (2016a). The term "surface seawater" used in this study defines a sample collected at less than 10 m depth. We also focused on the Japanese government's monitoring data at Tomari (42.98-43.17°N, 140.21-140.30°E), Aomori (41.13-41.22°N, 141.50-141.67°E), Niigata (37.62-38.10°N, 138.38-138.84°E), Ishikawa (36.87-37.29°N, 136.43-136.47°E), Fukui (35.75-36.09°N, 135.50-135.83°E), Shimane (35.67-35.80°N, 132.87-133.2°E), Saga (33.57-33.62°N, 129.73-129.98°E), and Kagoshima (31.58-31.93°N, 130.02-130.15°E) (Marine Ecology Research Institute, 2011, 2012, 2013, 2014, 2015, 2016) (Fig. 1). These measurements were taken once a year (from the middle of May to early June). Near the Aomori sites, offshore monitoring was also conducted twice a year (May and October) at the Rokkasho Reprocessing Plant (39.5-41.4°N, 141.5-142.3°E). Seawater was sampled from 0-664 m with different depths at each monitoring site. Monitoring data (304-01(33.0°N, 127.7°E), 105-11(37.3°N, 131.3°E)) from the Korean government was also used in this analy-

sis (Korea Institute of Nuclear Safety, 2011, 2012, 2013, 2014, 2015, 2016) (Fig. 1). At these monitoring sites, the surface seawater (0 m) measurements were taken four times (February, April, August, October) a year.

- There are also many other ways to gain insights into transport processes in the ocean. For surface pathways, one could be looking at satellite data (SST patterns and SSH data to derive geostrophic currents). For subsurface pathways, one could examine the subsurface dispersal of ARGO floats in the region and exploit numerical models. The latter includes (1) exploring the outputs of Eulerian mesoscale models (looking at the list of references, some outputs may be sourced by the authors themselves; the Japanese modelling group at the Earth-Simulator possibly run routinely models in the region; there are also some outputs publicly available, for instance on http://marine.copernicus.eu/services-portfolio/access-to- products/) as well as (2) performing ad-hoc Lagrangian experiments to unveil pathways (see also some references provided below). Altogether, these additional analyses could really help to grasp the three-dimensional structure of the flow and could provide further evidences to support their conclusions.

A. Data analysis by using satellite data, ARGO, model simulation, lagrangian experiments give us very useful information to invesrigate the transport of radiocaesium in the seawater. Authors recognized the importance of these data to interprete the transport as well as spatial and temporal distributions of FNPP1-radiocaesium. However, the evaluation of radiocaesium propagation from interior to the surface seawater did not obtain by using the satellite data. Although model simulation is very useful to interpret the transport and distribution of radiocasium, these also include the uncertainty. It is better to investigate the spatial and temporal vatiation of radiocaesium based on the measurement results at first stage. This matter is also suggested by the reviewer 1. The authors analyzed the radiocesium distributions by using only the observation data. The transport of radiocaesium by using the numerous data such as Japan Coastal Ocean Predicability Experiment (JCOPE) will be investigated as next step.

Discussion: I found that the results could be discussed further. For instance, the authors assumed that atmospheric deposition occurred only over the Pacific but what if airborne radiocaesium also felt down over the Sea of Japan? A. Just after the FNPP1 accident, 137Cs activity concentrations were increased in comparison with those before the FNPP1 accident (about 1.5 Bqm-3) in the northern part of the Sea of Japan (north of 40°N). The higher 137Cs activity concentrations exceed the concentrations before the FNPP1 accident was obsevred in the region north of 40 degree. In Fig. 1, 137Cs activity concentartions measured in 2011 were shown in color circles.

(Page 2, Line 16-21) In the northeastern part of the SOJ, atmospherically deposited radiocaesium were caused to approximately 1-2 times higher activity concentrations of radiocaesium in May 2011 (1.5-2.8 Bqm-3) in northeastern SOJ than those of before the FNPP1 accident (∼1.5 Bqm-3) (Fig. 1). By the end of 2011, the 137Cs activity concentrations in the northeastern part of the SOJ had rapidly decreased to almost the same levels as those before the FNPP1 accident (Inoue et al., 2012). In fact, the 137Cs activity concentration in surface water at an SOJ coastal monitoring site after July 2011 was almost the same as the pre-FNPP1 accident level (http://radioactivity.nsr.go.jp/ja/list/195/list-8.html).

- They conclude with a transport pathway crossing a well-established oceanic front : what kind of physical processes (front destabilization? Mesoscale processes? Etc. . .) could explain this route? What about discussing the effect of diffusive mixing?... A recent publication (Sania et al. PNAS 2017) suggests that continuous submarine groundwater discharge could also contribute to the radioactive elements measured in Japanese coastal waters; this process could also occur on the western shores. . .etc... A. In this study, we focus on the description of the spatial and temporal distributions of radiocaesium activity concentrations, FNPP1 derived radiocaesium activity concentrations, and 134Cs/137Cs activity ratio. In the revised manuscript, the authors discuss about the advection and vertical mixing of FNPP1-137Cs in the SOJ. As for the Sania et al.(2017), the unexepcted high activity concentrations of 137Cs (up to 23000 Bqm-3)

[revised manuscript text omitted]

warm subtropical waters and southward cold subarctic waters form the Polar Front meet at approximately 40°N, and the SOJ is largely divided into two regions (Prants et al., 2017). Rossi, V.; Van Sebille, E.; Sen Gupta, A.; GarcÌğon, V.; England, M. H. Multi-decadal projections of surface and interior pathways of the Fukushima Cesium-137 radioactive plume. Deep Sea Res., Part I 2013, 80, 37−46. A. This paper was re-submitted as Âń Corridendum to Âń Multi-decadal projections of surface and interor pathways of the Fukushima cesium-137 radioactive plume Âż (Deep Sea Reserach 1, 93, (2014), 162-164). In orther to describe the model simulation results, the author added Rossi et al. (2014) and Tsubono et al. (2016). (Page 1, Line 33-34) Ocean circulated models captured that the FNPP1-derived radiocaesium were transported in the North Pacific Ocean with advecting and diluting (Tsubono et al., 2016; Rossi et al., 2014).

– Some figures are not necessarily well chosen and some are not very informative due to their poor content and/or low quality and low visual rendering. This is especially true for fig. 2 (e.g. grey and red data points and error bars are not readable) A. Almost all Figures were rewrite. As for Fig.2, Authors think that the longtime variation of 137Cs actiity concentrations in the SOJ was necessary to discuss the enhanced activity concentrations of 137Cs after the FNPP1 accident. In order to show clearly the activity concentrations of 137Cs, the error bars of each sample was removed in this Figure. – fig. 3 (panels d, e and f: cut the x-axis in mid-2016 since there is no data afterwards); A. The period in the x-axis range for Fig 3d, e, f were modified and these were mover to Figure 4.

– fig. 5 (subplots are not numbered; what does black color means in panel 5c and 6c?); A. Fig. 5 was removed from the revised manuscript.

– the scatters in fig. 9 are not clear to me, please clarify. A. Before the FNPP1 accident, 137Cs were reelased from the large scale nuclear bomb experiment. After the FNPP1 accideint, 137Cs were derived from two sources, global fallout and FNPP1 accident. Because of shorter lifetime, 134Cs (T1/2=2.06 year) was only derived from

the FNPP1 accident. Therefore, the 134Cs/137Cs activity ratio is used as a useful traser to investigate the transport of radiocaesium derived from the FNPP1. It tends to the positive relationship between the 137Cs activity concentrations above 1.5 Bq m-3, which is the 137Cs activity concentrations before the FNPP1 accident, and the 134Cs/137Cs activity ratio. These relations in the SOJ suggests the contribution of 137Cs originaed from the FNPP1 accident. However, this relationship might be unclear for the reviewers. Therefore, this Figure was removed in the revied manuscript.

Please also note the supplement to this comment:
https://www.ocean-sci-discuss.net/os-2017-90/os-2017-90-AC2-supplement.pdf

---

## Referee Report (RR1)

**2nd Review of the manuscript OS-2017-90 entitled "Rapid transport of FNPP1-derived radiocaesium from subtropical mode water in the western North Pacific Ocean to the Sea of Japan" by Inomata et al.**

I thank the authors for their consideration of my previous comments. I acknowledge that they have made some efforts to improve the paper and to take into account some of my comments. However, I think the revised manuscript is not yet acceptable, it still contains mistakes and inconsistencies. I also found that some important results are kind of lost within "vague" discussion. I recommend minor revision; please be careful, extensive, rigorous and to the point this time.

**Major points:**

– The authors may have improved the English of the revised manuscript but not of the response letter which contains numerous mistakes and is, again, very difficult to read and understand. Some co-authors of this paper have extensive experience in publishing scientific papers in English; I am puzzled by the fact they have co-signed such English text (e.g. did they really read that paper and that response letter at least once?).

– I am convinced by the authors that the Cs137 signal in the SOJ first originates at the subsurface (e.g. 140 and 200 m) and then propagate to the surface, thus supporting their hypothesis of "mode water transport". However, I am not convinced by the first entrance through the Tsushima strait: indeed, on Fig. 3 c/d and Fig. 5 a/b, one can see that the subsurface peak of Cs137 occurred first (mid 2013) at station 105-11 (within the SOJ) and 1 year later at station 304-01 (ECS, e.g. upstream). Why?

– The "blob" of high Cs137 (8-12 Bq/m3) reported on Fig. 6a, b: could it be related to "exceptional sampling sites"; e.g. sampling within the core of an eddy that transport Cs137? (e.g. Prants etal. 2017 Ocean Sci. Discuss., doi:10.5194/os-2016-103, 2017).

– Please read better section 3.3. "Vertical distribution of Cs-137 and interior pathways" of Rossi et al. 2013 and please summarize their findings better in your introduction. The corrigendum concerns only the concentrations simulated (1 order of magnitude lower than reported), not the validity of the interior (and surface) pathways and subduction process.

– Figures are still of poor quality. For instance, labels have been misplaced in Fig. 7; labels are not readable on Fig. 8; etc…

– Fig. 7: those latitude-time plots are usually called "Hovmoller diagram".

– Title of section 3.2 (p 7): upstream sites are SJPN and ECS, right?

– Reference incomplete p 10 line 41.

– Abstract line 16-18: a sentence was repeated.

---

## Author Response (AR2)

**Dear Prof. Oka**

**Thank you very much for your review and comments. I modified the manuscript based on your comments and suggestions as well as other reviewer's comments.**

1. While the original manuscript insisted that the observed transport of radiocaesium suggests a new pathway of STMW circulation, the revised manuscript is more descriptive, focusing mainly on the radiocaesium transport from the East China Sea to the Japan Sea. I felt that this revision was adequate and suggest further revision to concentrate on the ECS to JS transport processes, because the authors are not showing any analysis nor discussion on the transport processes from the North Pacific to the ECS, particularly in Sec. 3.3.

   *The modified part was as follows:*

   [P6 L12-16]
   *We describe the possible pathway of FNPP1-$^{137}$Cs from the Pacific Ocean to the SOJ by ECS. Taking into account the physical ocean circulation, FNPP1-$^{137}$Cs in the STMW would be transported to southwestward by the westward undercurrent as reported by Oka (2009). The STMW containing $^{137}$Cs would reach the western boundary at lower latitudes, is entrained into the Kuroshio (Oka and Qiu, 2012), and is northward transported to the west of Kyushu by the TWC bifurcated from the Kuroshio.*
   *[P7 L 1-6]*
   *There were found several signatures that (i) The maximum $^{137}$Cs activity concentrations were observed in the subsurface layer in comparison with those in the surface seawater, (ii) This higher $^{137}$Cs activity concentrations were observed in the STMW based on the potential density data, (iii) In the ECS, the potential density in $^{137}$Cs existed seawater was almost same value those in the STMW, suggesting the signature of FNPP1-$^{137}$Cs transport into the ECS from the STMW in the NPSJ, (iv) The $^{137}$Cs activity concentrations in the northern ECS (>30°N) were higher than those in the southern ECS (<30°N) (Fig. SI12). In this study, however, the data was not enough to clarify the transport route in more detail.*

2. The schematic in Fig. 8 still needs to be revised. The Kuroshio path is too much simplified. Its pathway from the ECS to the NP via the Tokara Strait is not expressed at all. STMW is formed as deep winter mixed layers north of ~28N and is subducted southward across the mixed layer front at ~28N. Then, it is transport

westward and reaches the western boundary at ~25N or further south (e.g., Oka, 2009).

A. Fig. 8 was modified as shown. The Kuroshio current path was shown in more detail. In added to this, I added the STMW (formation region and transport region), CMW, and TRCMW.

[Figure]

3. As I commented to the original manuscript, I believe that STMW with radiocaesium does NOT "enter the ECS from the region between Kagoshima and Tokara" (p.7, l.1415) as shown by blue arrows in Fig. 8. If the authors insist so, please show the evidence.

A. Because our data was very limited, the description of the entrance region of FNPP1-$^{137}$Cs would be overstatement. The last sentence in Section 4.1 "FNPP1-radiocaesium might have entered the ECS from the region between Kagoshima and Tokara" was deleted.

[P6 L12-16]

*We describe the possible pathway of FNPP1-$^{137}$Cs from the Pacific Ocean to the SOJ by ECS. Taking into account the physical ocean circulation, FNPP1-$^{137}$Cs in the STMW would be transported to southwestward by the westward undercurrent as reported by Oka (2009). The STMW containing $^{137}$Cs would reach the western boundary at lower latitudes, is entrained into the Kuroshio (Oka and Qiu, 2012), and is northward transported to the west of Kyushu by the TWC bifurcated from the Kuroshio.*

Other comments:

1. Title: What does "rapid" mean? Given the timescale of 2-3 years for STMW circulation shown by previous studies, I think the time period for radiocaesium transport shown by the present study is reasonable but not "rapid".

A. The term of "rapid" was deleted from the title. The time scale of FNPP1-$^{137}$Cs transport from ECS to SOJ was consistent with those of nutrient species (e.g., Kodate et al., 2016).

The title was modified as follows.

*[P1. L1-2]*: *Transport of FNPP1-derived radiocaesium from subtropical mode water in the western North Pacific Ocean to the Sea of Japan*

2. P.1, l.16-18, "A time lag of the maximum .. in the SOJ.": I do not understand what each sentence mean and do not understand the relation between the two sentences.
A. This sentence was modified as follows.
*[P 1, L 15-21]*
*In the East China Sea (ECS), the clear increase in the $^{137}$Cs activity concentration started at 140 meters depth, density=25.2 kgm$^{-3}$, in April 2013, and propagated to the surface layers at approximately 0-50 meters depth, and showing a maximum in 2015 and decreasing in the following years. In the ECS, the maximum Fukushima-radiocaesium activity concentration in surface water was observed in 2014/2015, while it in the SOJ was observed in 2015/2016. Therefore, the propagation of Fukushima-derived radiocaesium in the surface seawater was approximately one year from ECS into the SOJ.*

3. p.2, l.31-32, "The SOJ is connected to the Pacific Ocean at its southwest through the Tsushima Straits": This is incorrect. The SOJ is connected to the East China Sea.

A. This description was unsuitableness. This sentence was also modified as follows.

*[P 2, L 30-31]*

*The SOJ is connected to the East China Sea at its southwest through the Tsushima Straits and connected to the North Pacific Ocean at its northeast through the Tsugaru Straits.*

4. p.2, l.33, "this current splits into three paths.": Please add references

A. There are several currents after entering the Sea of Japan such as East Korean Warm Current, North Korean Cold Current, the first and second branches of Tsushima Warm Current. Although several current in the SOJ, Tsushima Warm Current flows to the northeast through the Tsushima Strait from the ECS and mainly separated into two (west of TWC and east of TWC). Therefore, the description was modified as follows.

This part was modified as follows.

*[P 2, L 29-39]*

*The SOJ is located between the Eurasian continent and the Japanese archipelago. The area is 1008000 km$^2$, and the mean depth is 1667 m (Menard and Smith, 1966). The SOJ is connected to the East China Sea at its southwest through the Tsushima Straits and connected to the North Pacific Ocean at its northeast through the Tsugaru Straits. Warm and saline seawater passes through the Tsushima Strait as the Tsushima Warm Current (TWC), and this current splits into mainly two paths: One is the nearshore current along the west coast of Honshu Island, Japan, and the seawater passes through the Tsugaru Straits and enters to the Pacific Ocean again. The seawater transported to the northward passes thought the Soya Strait and connected to the Okhotsk Sea. Another current flows north of the Korean Peninsula. This current meets the North Korean Cold Current, which is the prolongation of the Liman Cold Current. Therefore, this northward warm subtropical waters and southward cold subarctic waters form the Polar Front meet at approximately 40°N, and the SOJ is largely divided into two regions (Prants et al., 2015). The increased FNPP1-$^{137}$Cs radioactivity concentrations were observed in the west coast of Honshu island (Aoyama et al., 2017).*

5. In the last paragraphs of Sec. 1, the authors need to explain more clearly what advances from the preliminary study of Aoyama et al. (2017) the authors are going to show in this study.

A. Based on the preliminary results, Aoyama et al. (2017) reported the increased of $^{137}$Cs activity concentrations in the surface seawater around the Japanese islands and ECS in winter 2015/2016. In this study, the author described the temporal variation of $^{137}$Cs

activities after the 1960's in the SOJ, and estimate the FNPP1-derived $^{137}$Cs activity concentrations. The author also investigated the vertical distributions of the $^{137}$Cs activity concentrations. The authors also discussed about FNPP1-$^{137}$Cs transport route from the NPSJ to the SOJ, estimate of transport amount with uncertainty of FNPP1-$^{137}$Cs into the SOJ as well as return amount to North Pacific Ocean from 2012 to 2016.

The modified part was as follows;
*[P 2, L 40~ P3 L3]*

*Aoyama et al. (2017) reported the increased of $^{137}$Cs activity concentrations in the surface seawater around the Japanese islands and ECS in winter 2015/2016 as a preliminary result. This research was investigated the temporal variation of $^{137}$Cs activities after the 1960's in the SOJ, and estimated the FNPP1-derived $^{137}$Cs activity concentrations after the year of 2011. We also investigated the vertical distributions of the $^{137}$Cs activity concentrations in the NPSJ, ECS, and SOJ. We also discussed about FNPP1-$^{137}$Cs transport route from the North Pacific South of Japan (NPSJ) to the SOJ, estimation of transport amount with uncertainty of FNPP1-$^{137}$Cs into the SOJ as well as return amount of FNPP1-$^{137}$Cs into the North Pacific Ocean during the period from 2012 to 2016.*

6. p.5, l.20: The wording "the southern Japan (SJPN)" does not make sense. I would call the region "North Pacific south of Japan (NPSJ)".
A. The term "the southern Japan (SJPN)" was modified to "*North Pacific south of Japan (NPSJ)*".

The modified part was as follows;
*[P 5, L 22]*
*Propagation of radiocaesium from the upstream region (NPSJ and ECS) to the downstream region (the SOJ)*

7. p.5, l.34-35, "Fig. 7 displays the latitude-time cross-sections … along the eastern TWC.": How did the authors distinguish the eastern and western TWC? Please explain.
A. At first, the cross sections along the western TWC (off Korean Peninsula) and the eastern TWC (along the Honshu islands in Japan) was depicted in Fig. 7. However, the data along with the western TWC was very limited and omitted in the revised version. The term "latitude-time cross-sections" was also modified as "Hovmoller diagrams".

The modified part was as follows.

*[P 5, L 38-39] Fig. 7 displays the Hovmoller diagrams of the $^{137}$Cs activity concentrations at potential temperatures of 25.2±0.5 kgm$^{-3}$ along the TWC in the ECS and coastal site of eastern SOJ.*

8. p.6, l.6: What does "the other layers" mean?

A. As described above, the Hovmoller diagrams of the $^{137}$Cs activity concentrations along the western TWC (off Korean Peninsula) was deleted. Then the description (P6, L5-10) was also deleted.

Revisesd manuscript OS-2017-90 entitled "Transport of FNPP1- derived radiocaesium from subtropical mode water in the western North Pacific Ocean to the Sea of Japan" by Inomata et al.

*The authors appreciate to the 2nd review and positive comments. We modified the manuscript based on your comments.*

**Major points:**

– The authors may have improved the English of the revised manuscript but not of the response letter which contains numerous mistakes and is, again, very difficult to read and understand. Some co-authors of this paper have extensive experience in publishing scientific papers in English; I am puzzled by the fact they have co-signed such English text (e.g. did they really read that paper and that response letter at least once?).

A. The English was carefuly checked and modified in the revised version and response letter.

– I am convinced by the authors that the Cs137 signal in the SOJ first originates at the subsurface (e.g. 140 and 200 m) and then propagate to the surface, thus supporting their hypothesis of "mode water transport". However, I am not convinced by the first entrance through the Tsushima strait: indeed, on Fig. 3 c/d and Fig. 5 a/b, one can see that the subsurface peak of Cs137 occurred first (mid 2013) at station 105-11 (within the SOJ) and 1 year later at station 304-01 (ECS, e.g. upstream). Why?

A. X-axis in Figure 5a and b show different periods. The aithour apologized that this might be confusing the reviewer. The x-axis in Figure 5 a and 5b was modified.
The increase of $^{137}$Cs activity concentrations was observed in the year of 2013 in the stations of 314-01 (140 m depth) and 105-11 (200 m depth). The $^{137}$Cs activity concentrations in the year of 2013 was almost same in the stations of 314-01 (2.1±0.16, 2.1±0.19 Bqm$^{-3}$) and 105-11 (2.4±0.18 Bqm$^{-3}$) (see Figure 5) or slightly higer in the station 105-11. The increase of $^{137}$Cs activity concentrations at subsurface layer occurred in 2013 and this was approximately 1 year earlier than that in the shallower layers. Difference of $^{137}$Cs activity concentrations among the two stations would strongly reflect whether the seawater locate the center of mode water transport or not. As shown in Fig. 6(e) and (f), the $^{137}$Cs existed in the denser layer than those in the subtropical mode water, resulting with the cooling in the surface layer by southward transported colder seawater after the sea water was transported from the Tsushima Strait. This effect would also influence the slight difference of $^{137}$Cs activity concentrations in the stations of 314-01 and 105-11.

*The modified parts were as follows ;*
*[P5, L 9-12] : Fig. 5 show the temporal variation in the $^{137}$Cs activity concentrations at different depths at station 314-01 in the ECS (Fig. 5a; 0-140 m) and station 105-11 in the SOJ (Fig. 5b; 0-2000 m). The increase in the $^{137}$Cs activity concentrations at the subsurface layer (140 m at the station 314-01, 200 m at the station 105-11) occurred in 2013 and this was approximately 1 year earlier than that in the shallower layers.*

*[P 7, L 30-38]*

*As shown in Fig. 6e, the $^{137}$Cs activity concentrations at station 105-11, located along the western TWC in the SOJ, were maximum at the surface and gradually decreased with increasing depth. This vertical distribution is different from those in the NPSJ (Fig. 6a) and ECS (Fig. 6c). Particularly, the subsurface*

*peak observed in the NPSJ and ECS did not appear at station 105-11. At station 105-11, most of the $^{137}$Cs existed in seawater with a potential density of 25.7-27.3 kg m$^{-3}$ (Fig. 6f), which was located in a higher potential density layer than that in the NPSJ and ECS. A similar vertical distribution was also observed at the western coast of the Japanese Islands along the eastern TWC (Fig. SI11). These distributions were due to cooling in the surface layer after water was transported from the Tsushima Strait. Physical processes such as the convergence and subduction of surface water inside the eddies are important mechanisms of downward transport of radiocaesium (Miyao et al., 1998; Budyansky et al., 2015).*

– The "blob" of high Cs137 (8-12 Bq/m$^3$) reported on Fig. 6a, b: could it be related to "exceptional sampling sites"; e.g. sampling within the core of an eddy that transport Cs137? (e.g. Prants etal. 2017 Ocean Sci. Discuss., doi:10.5194/os-2016-103, 2017).

    A. Prants et al. (2017) identified the high activity of radioactivity in the mesoscale eddies by lagrangian methodology. However, the high Cs137 (8-12 Bq/m$^3$) reported on Fig. 6a, 6b were observed in the region south of Kuroshio based on hte temperature profile. Furthremore, based on the analysis by AVISO, these data was obseverd in the region far from the eddies. I think that these data did not show the signature of direct transport across Kuroshio by the eddy effect.

– Please read better section 3.3. "Vertical distribution of Cs-137 and interior pathways" of Rossi et al. 2013 and please summarize their findings better in your introduction. The corrigendum concerns only the concentrations simulated (1 order of magnitude lower than reported), not the validity of the interior (and surface) pathways and subduction process.

    A. The author described the penetration of FNPP1-derived $^{137}$Cs in the ocean interior by numerical model simulations (Rossi et al., 2013).

    *The modified parts were as follows ;*
*[P 2, L 7-11]; Rossi et al. (2013) reported that simulated FNPP1-derived $^{137}$Cs were penetrated into the ocean interior by intense subduction and vertical mixing on winter season around the formation regions of mode waters as described by Oka and Qiu (2012). In their results, the higher $^{137}$Cs activity concentrations associated by the subduction was also reproduced in the STMW, Denser, and Lighter CMW.* About 43% of FNPP1-$^{137}$Cs was transported downward below the mixed layer by eddy processes (Kamidaira et al., 2018).

– Figures are still of poor quality. For instance, labels have been misplaced in Fig. 7; labels are not readable on Fig. 8; etc…

– Fig. 7: those latitude-time plots are usually called "Hovmoller diagram".

    In the revised version, the author selected only one Figure 7a because the data in Figure 7b and c were very limited. The quality of new Figure 7 is improved. Figure 8 is re-depcited by the suggesttions of reviwer 1. I also take into account to be higher quality for all Figures.

    A. The plots were modified as "Hovmoller diagram".

    *The modified parts were as follows ;*

*[P 5, L 38-39]; Fig. 7 displays the Hovmoller diagrams of the $^{137}$Cs activity concentrations at potential temperatures of 25.2±0.5 kgm$^{-3}$ along the TWC in the ECS and coastal site of eastern SOJ.*

*Caption ; Figure 7: Hovmoller diagrams of the $^{137}$Cs activity concentrations at a potential density along with an eastern TWC at a potential density of 25.2 ±0.5 kg m$^{-3}$. The ECS stations described on the x-axis are Kagoshima and Saga stations. Southwestern SOJ includes the monitoring stations Shimane, Fukui, Ishikawa, and Niigata. The northwestern SOJ includes the monitoring stations Aomori and Tomari. Color indicates the $^{137}$Cs activity concentrations (Bq m$^{-3}$).*

–   Title of section 3.2 (p 7): upstream sites are SJPN and ECS, right?

A. The author considered the NPSJ (modified from SJPN accoding to the reviewer 1's comment) and ECS are upstream sites.

*The modified part was as follows;*

[revised manuscript text omitted]

---

## Author Response (AR4)

Dear Editor

Thank you very much for your comments.

We have rewritten the purpose of this study in the introduction.

Furthermore, we had native speakers of English proofread our English writing once more.

Please kind treatment.

Best regards,

Yayoi Inomata

For Prof. Oka

Thank you very much for your review of our manuscript. We were re-write the purpose of this study in the last paragraph in the introduction. Furthermore, we have checked the English by the native speakers.

Thank you very much again.

1. "at the end of Sec. 1, the authors need to clearly mention the purpose of this manuscript"

Answer

The purpose of the manuscript was re-written in the last paragraph in section 1 (Introduction).

P2. L36-40: The purposes of this study were to 1) investigate the spatiotemporal variations of activity concentrations in the SOJ of 137Cs released as a result of the FNPP1 accident, 2) investigate the processes responsible for transport of radiocaesium from the North Pacific Ocean to the SOJ through the East China Sea, and 3) estimate the amount of FNPP1-derived 137Cs transported into the SOJ through the Tsushima Strait as well as the amount of FNPP1-derived 137Cs returned into the North Pacific Ocean through the Tsugaru Strait during 2012–2016 and the uncertainties of these estimates.

2. "There are still numerous grammar mistakes in the manuscript, but I believe the journal will correct them."

As for the English, we had native speakers of English proofread our English writing.